# Understanding the Evolution of Linear Regions in Deep Reinforcement Learning

**Setareh Cohan**
Department of Computer Science
University of British Columbia
setarehc@cs.ubc.ca

**Nam Hee Kim**
Department of Computer Science
Aalto University
namhee.kim@aalto.fi

**David Rolnick**
School of Computer Science
McGill University
drolnick@cs.mcgill.ca

**Michiel van de Panne**
Department of Computer Science
University of British Columbia
van@cs.ubc.ca

## Abstract

Policies produced by deep reinforcement learning are typically characterised by their learning curves, but they remain poorly understood in many other respects. ReLU-based policies result in a partitioning of the input space into piecewise linear regions. We seek to understand how observed region counts and their densities evolve during deep reinforcement learning using empirical results that span a range of continuous control tasks and policy network dimensions. Intuitively, we may expect that during training, the region density increases in the areas that are frequently visited by the policy, thereby affording fine-grained control. We use recent theoretical and empirical results for the linear regions induced by neural networks in supervised learning settings for grounding and comparison of our results. Empirically, we find that the region density increases only moderately throughout training, as measured along fixed trajectories coming from the final policy. However, the trajectories themselves also increase in length during training, and thus the region densities decrease as seen from the perspective of the current trajectory. Our findings suggest that the complexity of deep reinforcement learning policies does not principally emerge from a significant growth in the complexity of functions observed on-and-around trajectories of the policy.

## 1 Introduction

Deep reinforcement learning (RL) utilizes neural networks to represent the policy and to train this network to optimize an objective, typically the expected value of time-discounted future rewards. Deep RL algorithms have been successfully applied to diverse applications including robotics, challenging games, and an increasing number of real-world decision-and-control problems [François-Lavet et al., 2018]. For a given choice of task, RL algorithm, and policy network configuration, the performance is commonly characterised via the learning curves, which provide insight into the learning efficiency and the final performance. However, little has been done to understand the detailed structure of the state-to-action mappings induced by the control policies and how these evolve over time.

In this work, we aim to further understand deep feed-forward neural network policies that use rectified linear activation functions (rectifier linear units or ReLUs). ReLUs [Nair and Hinton, 2010] are among the most popular choices of activation functions due to their practical successes [Montúfar et al., 2014]. For RL, these activations induce a piecewise linear mapping from states to actions,

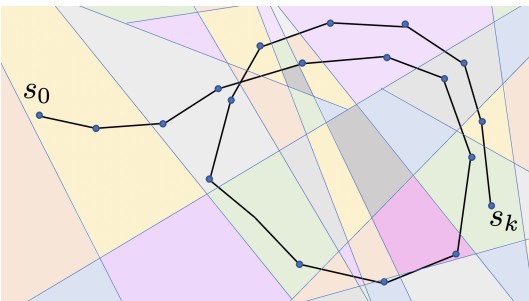

Figure 1: Schematic illustration of a trajectory traversing the piecewise linear regions in the policy state space. $S_0$ and $S_k$ indicate the initial and final states of the trajectory.

where the input space, i.e., the state space, is divided into distinct *linear regions*, where within each region, the actions are a linear function of the states. Note that the regions are formally an affine function of the input, due to the constant-valued bias terms. For simplicity and convenience, these are more commonly described as linear regions. Figure 1 provides a schematic illustration of these regions, along with a policy trajectory.

The number of distinct regions into which the input space is divided is a natural measure of network expressivity. Learned functions with many linear regions have the capacity to build complex and flexible decision boundaries. Thus, the problem of counting the number of linear regions has been extensively studied in recent literature [Montúfar et al., 2014, Raghu et al., 2017, Hanin and Rolnick, 2019a, Serra et al., 2018]. While the maximum number of regions is exponential with respect to network depth [Montúfar et al., 2014], recent work has demonstrated that the number of regions is instead typically proportional to the number of neurons [Hanin and Rolnick, 2019a].

For RL, we are interested in the local granularity (density) of linear regions along trajectories arising from the policy. Fine-grained regions afford fine-grained control, and thus we may hypothesize that region density *increases* in regions frequently visited by the policy, in order to afford better control. Recent work in supervised learning of image classification is inconsistent with regard to findings about the region density seen in the vicinity of data points, with some reporting a decrease [Novak et al., 2018] to provide better generalization and robustness to perturbation, and others not [Hanin and Rolnick, 2019a]. For the RL setting, we note that counting regions visited along an episode trajectory arguably provides a meaningful and task-grounded measurement in contrast to line segments and ellipses that pass through points randomly sampled from training data, which have been used in the prior works mentioned above. We further note that piecewise-affine control strategies are commonly designed into control systems, e.g., via gain scheduling. Understanding how these regions are designed and distributed by deep RL thus helps establish bridges with these existing methods.

To the best of our knowledge, our work is the first to investigate the structure and evolution of linear regions of ReLU-based deep RL policies in detail. We seek to answer several basic empirical questions :

**Q1** Do findings for network expressivity, originally developed in supervised learning settings, apply to RL policies? How are the region densities affected by the policy network configuration? Do deeper policy networks result in finer-grained regions and hence an increased expressivity?

**Q2** How do the linear regions of a policy evolve during training? Do we see a significantly greater density of regions emerge along the areas of the state space frequently visited by the episodic trajectories, thereby allowing for finer-grained control? Do random-action trajectories see different densities?

The key results can be summarized as follows, for policies trained using proximal policy optimization (PPO) [Schulman et al., 2017], and evaluated on four different continuous control tasks. Q1: There is a general alignment with recent theoretical and empirical results for supervised learning settings. Region density is principally proportional to the number of neurons, with an additional small observed increase in density for deeper networks. Q2: Only a moderate increase of density is observed during training, as measured along fixed final-policy trajectories. Therefore, the complexity of a final learned

policy does *not* come principally from increased density on-and-around the optimal trajectories, which is a potentially surprising result. In contrast, as measured along the evolving current-policy trajectories, a decrease in region density is observed during training. Across all settings, we also observe that the region-transition count, as observed during fixed time-duration episodes, grows during training before converging to a plateau. However, the trajectory length, as measured in the input space, also grows towards a plateau, although not at the same rate, and this leads to variations in the mean region densities as observed along current trajectories during training.

## 2 Related Work

Understanding the expressivity of a neural network is fundamental to better understanding its operation. Several works study expressivity of deep neural networks with piecewise linear activations by counting their linear regions [Arora et al., 2016, Bianchini and Scarselli, 2014]. On the theoretical side, Pascanu et al. [2013] show that in the asymptotic limit of many hidden layers, deep ReLU networks are capable of separating their input space into exponentially more linear regions compared with shallow networks, despite using the same number of computational units. Following this, Montúfar et al. [2014] also explore the complexity of functions computable by deep feed-forward neural networks with ReLU and maxout activations, and provide tighter upper and lower bounds for the maximal number of linear regions. They show that the number of linear regions is polynomial in the network width and exponential with respect to the network depth. Furthermore, Raghu et al. [2017] improve the upper bound for ReLU networks by introducing a new set of expressivity measures and showing that expressivity grows exponentially with network depth for ReLU and Tanh activated neural networks. Serra et al. [2018] generalize these results by providing even tighter upper and lower bounds for the number of regions for ReLU networks and show that the maximal number of regions grows exponentially with depth when input dimension is sufficiently large.

Recent studies on the expressivity of neural network depth suggest that the actual impact of depth on expressivity is likely far below that of the theoretical maximum proposed by prior literature. Hanin and Rolnick [2019b] study the importance of depth on expressivity of neural networks in practice. They show that the average number of linear regions for ReLU networks at initialization is bounded by the number of neurons raised to the input dimension, and is independent of network depth. They also empirically show that this bound remains tight during training. Similarly, Hanin and Rolnick [2019a] find that the average distance to the boundary of the linear regions depends only on the number of neurons and not on the network depth – both at initialization and during training for supervised-learning tasks on ReLU networks. This strongly suggests that deeper networks do not necessarily learn more complex functions in comparison to shallow networks. Prior to this, a number of works have shown that the strength of deep learning may arise in part from a good match between deep architectures and current training procedures [Mhaskar and Poggio, 2016, Mhaskar et al., 2016, Zhang et al., 2021]. Notably, Ba and Caruana [2014] show that once deep networks are trained to perform a task successfully, their behavior can often be replicated by shallow networks, suggesting that the advantages of depth may be linked to easier learning.

Another line of work studies function complexity in terms of robustness to perturbations to the input. Sokolić et al. [2017] theoretically study the input-output Jacobian, which is a measure of robustness and also relates to generalization. Similarly, Zahavy et al. [2016b] propose a sensitivity measure in terms of adversarial robustness and provide theoretical and experimental insights on how it relates to generalization. Novak et al. [2018] also study robustness and sensitivity using the input-output Jacobian and the number of transitions along trajectories in the input space as measures of robustness. They show that neural networks trained for image classification tasks are more robust to input perturbations in the vicinity of the training data manifold, due to training points lying in regions of lower density. Several other recent works have also focused on proposing tight generalization bounds for neural networks [Bartlett et al., 2017, Dziugaite and Roy, 2017, Neyshabur et al., 2017].

There are a number of works that touch on understanding deep neural networks by finding general principles and patterns during training. Arpit et al. [2017] empirically show that deep networks prioritize learning simple patterns of the data during training. Xu et al. [2019] find a similar phenomenon in the case of 2-layer networks with Sigmoid activations. Rahaman et al. [2019] study deep ReLU activated networks through the lens of Fourier analysis and show that while deep neural networks can approximate arbitrary functions, they favor low-frequency ones, and thus, they exhibit a bias towards smooth functions. Samek et al. [2017] present two approaches in explaining predictions

of deep learning models in a classification task, with the first method computing the sensitivity of the prediction with respect to input perturbations and the second method that meaningfully decomposes the decision in terms of input variables.

While deep RL methods are widely used and extensively studied, few works focus on understanding the policy structure in detail. Zahavy et al. [2016a] propose a visualization method to interpret the agent's actions by describing the Markov Decision Process as a directed graph on a t-SNE map. They then suggest ways to interpret, debug and optimize deep neural network policies using the proposed visualization maps. Rupprecht et al. [2019] train a generative model over the state space of Atari games to visualize states which minimize or maximize given action probabilities. Luo et al. [2018] adapt three visualization techniques to the domain of image-based RL using convolutional neural networks in order to understand the decision making process of the RL agent.

## 3 Piecewise Linear Regions

Throughout this work, we consider RL policies based on *ReLU networks*. A ReLU network is a ReLU-activated feed-forward neural network, or a multi-layer perceptron (MLP) which can be formulated as a scalar function $f : \mathbb{R}^d \to \mathbb{R}^o$ defined by a neural network with $L$ hidden layers of width $d_1, ..., d_L$ and a $o$-dimensional output. Assuming the output to be 1-dimensional, and following the notation of Rahaman et al. [2019], we have

$$f(\mathbf{x}) = (T^{(L+1)} \circ \sigma \circ T^{(L)} \circ ... \circ \sigma \circ T^{(1)})(\mathbf{x}), \tag{1}$$

where $T^{(k)} : \mathbb{R}^{d_{k-1}} \to \mathbb{R}^{d_k}$ computes the weighted sum $T^{(k)}(\mathbf{x}) = W^{(k)}\mathbf{x} + \mathbf{b}^{(k)}$ for some weight matrix $W^{(k)}$ and bias vector $\mathbf{b}^{(k)}$. Here, $\sigma(\mathbf{u}) = \max(0, \mathbf{u})$ denotes the ReLU activation function acting element-wise on vector $\mathbf{u} = (u_1, ..., u_n)$.

Given the ReLU network $f$ from Equation 1, again following Rahaman et al. [2019], piecewise linearity can be explicitly written by

$$f(\mathbf{x}) = \sum_{r \in R} 1_{P_r}(\mathbf{x})(W_r\mathbf{x} + \mathbf{b}_r) \tag{2}$$

where $r$ is the index for the linear region $P_r$ and $1_{P_r}$ is the indicator function on $P_r$. The $1 \times d$ matrix $W_r$ is given by:

$$W_r = W^{(L+1)}W_r^{(L)}...W_r^{(1)} \tag{3}$$

where $W_r^{(k)}$ is obtained from the original weight $W^{(k)}$ by setting its $j^{\text{th}}$ column to zero whenever neuron $j$ of layer $k-1$ is inactive for $k > 1$.

## 4 Counting Linear Regions in RL Policies

To answer the key questions posed in the introduction, we need a method for counting the linear regions encountered through the episodic trajectories taken during RL. For each input $\mathbf{s}$, we encode each neuron of the policy network with a binary code of $0$ if its pre-activation is negative, and with a binary code $1$ otherwise. The linear region of the input $\mathbf{s}$ can thus be uniquely identified by the concatenation of binary codes of all the neurons in the network called an *activation pattern*. Figure 2 illustrates the activation pattern construction for a 2D input space.

Figure 1 provides a schematic illustration of the linear regions for a 2D state space, together with an example trajectory that consists of sequential states encountered by the policy, connected with straight-line segments. As seen in the figure, regions can be revisited, which leads to a distinction between the number of transitions between regions and the number of unique regions visited along a trajectory. We compute both of these metrics along episodic trajectories, as enabled by keeping a list

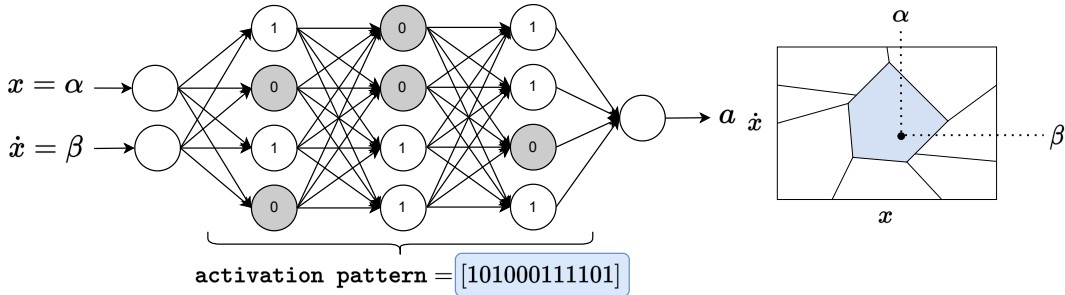

Figure 2: An overview of a ReLU-activated policy network and the binary labeling scheme of the linear regions. State $\mathbf{s} = [x, \dot{x}]$ is within the linear region uniquely identified by the activation pattern computed by concatenating the binary states of the activations of the policy network given input $\mathbf{s}$.

of the regions already visited at any point in time when processing a given episode trajectory. Our region-counting method also includes regions that are crossed by the straight-line segments between successive trajectory states, although the policy itself does not explicitly encounter these regions due to the discrete-time nature of typical RL environments.

To represent the $k$-th line segment of the episodic trajectories in the input space, we develop a parameterized line segment given by $s(u) = (1-u)s_{k-1} + us_k$, where $u \in [0, 1]$ and $s_{k-1}, s_k \in \mathbb{R}^d$ denote the endpoints of the line segment. We then calculate the exact number of the linear regions over this line segment by considering the hidden layers of the policy network one at a time, starting from input towards the output, and observe how each region can be split by the neurons. Starting with the first layer, we consider neurons one by one, and identify the point in the domain of $u$, if any, that induces a change in the binary labeling for that neuron. Each such point subdivides the domain of $u$ into two new regions where the pre-activation will be zero for one of them in the next layer. By maintaining a list of these regions and the linear functions defined over them, and whether their pre-activation vanishes in the next layer, we proceed to the neurons of the next layer. This process repeats for each of the regions resulting from the previous layer. We record the activation patterns of all the final regions, to track state visits during all segments of episode trajectories. In the end, for each trajectory $\tau$, we compute the total region transitions, $R_T(\tau)$, as well as the number of unique visited regions, $R_U(\tau)$, where $R_T(\tau) + 1 \geq R_U(\tau)$. We further compute the trajectory length in the input space, $L(\tau)$. This allows us to compute normalized region densities according to $\rho(\tau) = R_T(\tau)/(NL(\tau))$ where $N$ is the total number of neurons in the policy network, in accordance with Hanin and Rolnick [2019a].

We use the above metrics along two types of trajectories. First, we consider *fixed* trajectories, $\tau^*$ as sampled from the final fully-trained policy. Second, we consider *current* trajectories, $\tau$, as sampled from the current snapshot of the policy during training. Both of these trajectories offer informative views of the evolution of the policy. The former offers a direct picture of the linear-region density along a meaningful, fixed region of the state space, i.e., that of the final optimized policy. The latter offers a view of what the policy trajectories actually encounter during training. In order to better understand the evolution of $\tau$, we also track its length, $L(\tau)$, given that the density is determined by the region-transition count as well as the length of the trajectory. Figure 3 shows the evolution of linear regions and the types of trajectories $\tau^*$ and $\tau$ for a simple 2D toy environment, and a ReLU-activated policy network of depth 2 and width 8.

## 5  Experimental Results

We conduct our experiments on four continuous control tasks including HalfCheetah-v2, Walker-v2, Ant-v2, and Swimmer-v2 environments from the OpenAI gym benchmark suits [Brockman et al., 2016]. We use the Stable-Baselines3 implementations of the PPO algorithm [Schulman et al., 2017] in all of our experiments throughout this work [1]. We run each experiment with 5 different random seeds and the results show the mean and the standard deviation across the random seeds. The policy network architecture is an MLP with ReLU activations in all hidden layers in our experiments.

---

[1] Our code is available at `https://github.com/setarehc/deep_rl_regions`.

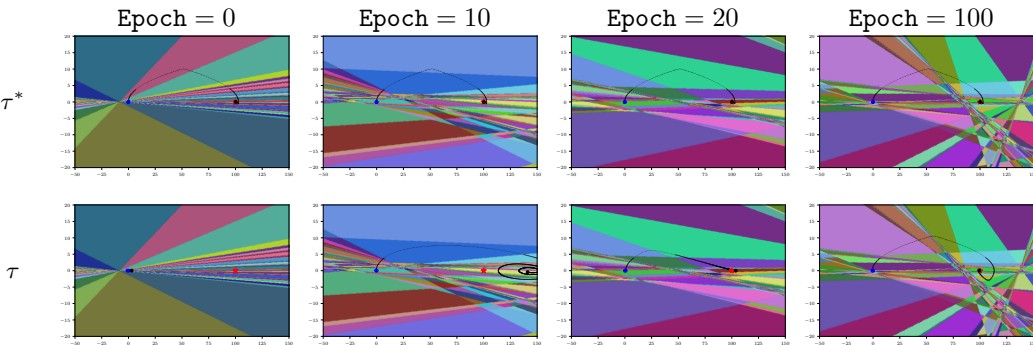

Figure 3: State space visualization with current ($\tau$) and fixed trajectories ($\tau^*$) of a ReLU-activated policy network with depth 2 and width 8 trained on a 2D toy environment during training. The first row, plots a fixed trajectory sampled from the final policy over the state space. The second row plots a trajectory sampled from the current snapshot of the policy. Columns indicate the number of training epochs. Epoch = 0 corresponds to the policy at initialization and Epoch = 100 corresponds to the fully-trained policy. In this toy environment, an agent starts at the origin indicated with a blue dot and the goal is for the agent to reach a target state located at $x = 100$. States are the position $x$ (horizontal axis) and velocity $\dot{x}$ (vertical axis) of the agent, and action is the acceleration.

We train 18 policy network configurations with $N \in \{32, 48, 64, 96, 128, 192\}$ neurons, widths $w \in \{8, 16, 32, 64\}$, and depths $d \in \{1, 2, 3, 4\}$. We use a fixed value function network structure of $(64, 64)$ in all of our experiments. Details of the network configurations used for the policy network are available in Appendix C. These network configurations are chosen such that the network is fully-capable of learning the task and achieves near state-of-the-art cumulative reward on the particular task it is trained on. This allows us to study the evolution of linear regions disentangled from properties such as trainability, which is outside the scope of this work. The table showing the range of the mean cumulative reward per task, across all network configurations can be found in Appendix B.

We adopt the network initialization and hyperparameters of PPO from StableBaselines3 [Raffin et al., 2021] and train our policy networks on 2M samples (i.e. 2M timesteps in the environment). This initialization scheme also matches the best-practice recommendations for on-policy RL, e.g., [Andrychowicz et al., 2020], although it differs from that used in Hanin and Rolnick [2019a] for theoretical and empirical linear region analysis. A detailed explanation of the initialization and choice of hyperparameters is available in Appendix D.

In what follows, we focus on the results for the HalfCheetah-v2 environment as a representative example. Similar results for the other environments are provided in Appendix K.

### 5.1 Evolution of the Density of Linear Regions during Training

We begin by examining the number of transitions encountered as we sweep along the fixed final-policy trajectory, $R_T(\tau^*)$, as shown in Figure 4(a). For clarity, we present results for only three policy network configurations. This plot shows a moderate and gradual increase of 50% in the number of region transitions observed in fixed-duration RL episodes during training, with larger policy networks having more transitions. Computing the region density requires dividing by number of neurons, $N$, and the trajectory length, $L(\tau^*)$, which is constant for the fixed final trajectory, $\tau^*$. The result is given in Figure 4(b) which shows the mean and standard deviation of the normalized region densities across policy networks with the same depth. This shows a result that is consistent across all policy network dimensions to within $\pm 30\%$, and reflects an identical moderate-and-gradual increase during training, regardless of the policy network configuration. Thus, in response to **Q2**, RL policies trained with PPO see a moderate increase in region density along the parts of the state space that are frequently visited by the policy. We later discuss the observed impact of policy-network depth (§5.2).

We next examine the number of transitions encountered as we sweep along the evolving set of current-policy trajectories, $R_T(\tau)$, shown in Figure 4(c). These begin near zero and then rapidly increase as the policy succeeds at exploration. Again, for clarity, we show a representative sample of

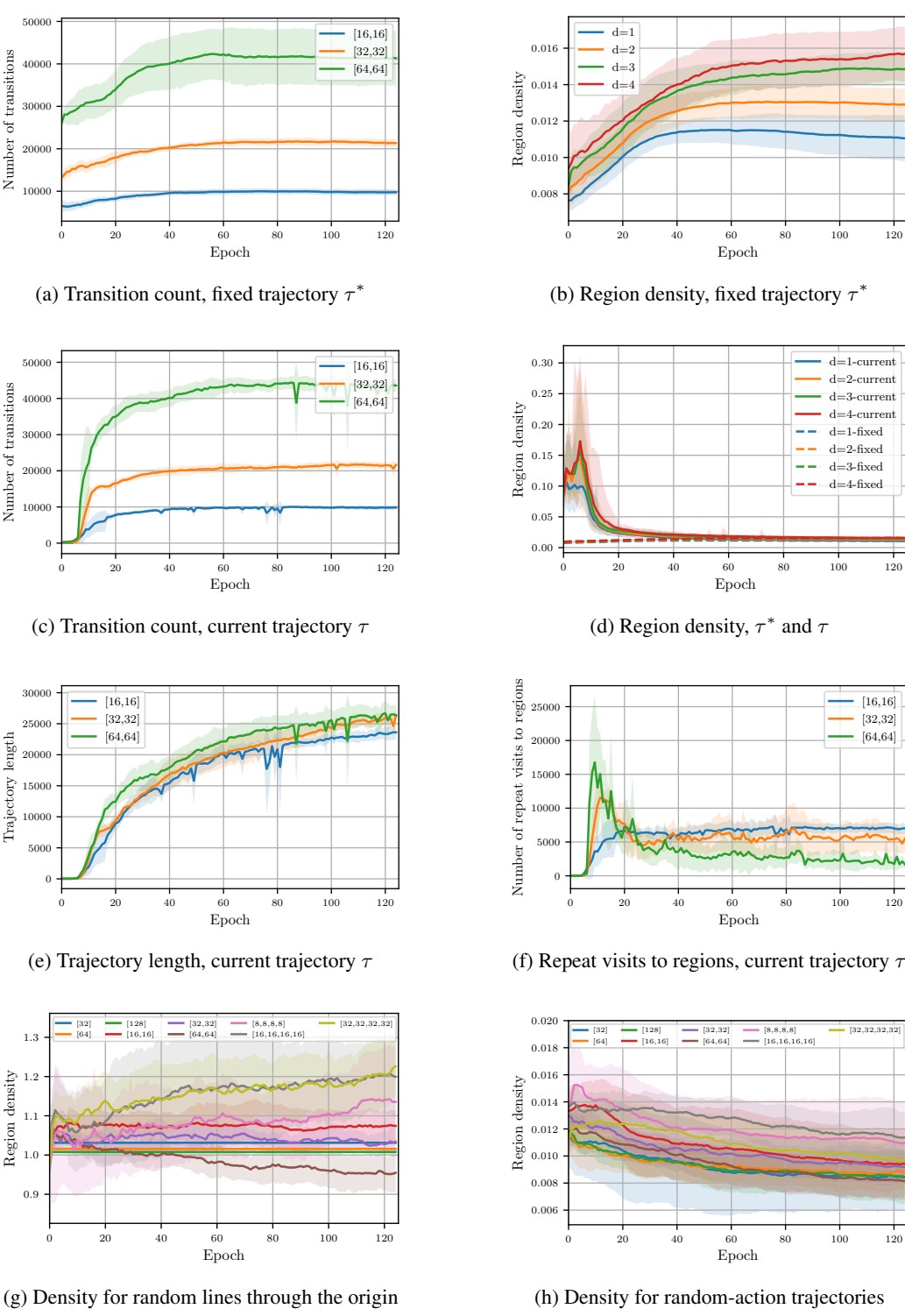

(a) Transition count, fixed trajectory $\tau^*$

(b) Region density, fixed trajectory $\tau^*$

(c) Transition count, current trajectory $\tau$

(d) Region density, $\tau^*$ and $\tau$

(e) Trajectory length, current trajectory $\tau$

(f) Repeat visits to regions, current trajectory $\tau$

(g) Density for random lines through the origin

(h) Density for random-action trajectories

Figure 4: Evolution of the number of transitions, linear region densities, and length of trajectories for HalfCheetah, during training. Results for the rest of the environments are provided in Appendix K. The ranges indicate the standard error across 5 random seeds. $[n_1, ..., n_d]$ in the legend corresponds to a network architecture with depth $d$ and $n_i$ neurons in each layer. Summary results are grouped by network depth, $d$.

three policy network dimensions here. The corresponding region densities are shown in Figure 4(d),

together with the fixed-region densities which are repeated from Figure 4(b) (see the bottom dashed curves). We speculate that the observed densities are higher early on due to the unstructured nature of early exploration and the form of network initialization, and then decrease over time as the trajectory lengths begin to increase, as shown in Figure 4(e).

We compute the number of repeat visits to regions as $R_T - R_U$, i.e., the region transition count minus the unique region count. In Figure 4(f), we track this for the current trajectory during training. We speculate that repeat visits are high early in training because of limited exploration. Repeat visits remain later in training because of the cyclic nature of the learned locomotion for HalfCheetah.

## 5.2 Impact of Policy Network Structure on Expressivity

Our normalized density computation assumes that the number of transitions scales linearly with the number of neurons, $N$. However, we empirically observe that there remains a moderate dependence on the network depth, as seen in Figure 4(b), with deeper networks resulting in moderately denser regions in the learned policies. We further examine the policy network expressivity using random infinite lines. Following Hanin and Rolnick [2019a], for each policy network, we sample 100 random lines by randomly sampling 100 points from the final trajectory $\tau^*$, and connecting each randomly sampled point to the origin of the policy input space. Figure 4(g) shows the mean and standard deviation of the normalized number of transitions, $R_T/N$, as computed along these infinite lines in random directions passing through the origin, over the 5 random seeds of each network configuration. These produce approximately uniform results throughout training, showing no overall changes in mean density in various directions. These results are also consistent with the findings of Hanin and Rolnick [2019a] as they show normalized region densities are close to 1 at initialization and remain roughly constant during training. We repeated the same experiment with 100 random lines through the mean point instead of the origin, by connecting each randomly sampled point to the mean of the 100 sampled points. The difference between the density over lines through the origin and through the mean point is negligible.

To further examine the behavior of the policy network on areas that it does not focus on exploring during training, we compute the region density along random trajectories created by taking random actions in the environment. For each policy network, we sample a set of 10 such random trajectories $\tau^R$ and compute the normalized region density over these. Figure 4(h) shows the mean and standard deviation of this metric across this set of 10 random trajectories, over the 5 random seeds of each network configuration. This shows a very moderate decrease during training, with fairly consistent densities for all policy network configurations. We speculate that the slight decrease in density over time arises from the trained policy no longer visiting the same regions as the random-action policy. Comparing the results of random trajectories with fixed and current (Figure 4(b) and Figure 4(d)), we can see that the normalized region density over random-action trajectories and a fixed final trajectory are within the same range. Also, although this metric is higher over current trajectories early on during training, it eventually decreases and converges to the same range of region densities as the fixed trajectory. In Appendix G we further directly visualize the regions of a policy network for HalfCheetah, as viewed via an embedding plane that either passes through three points sampled from the final fixed trajectory, or three points sampled from a random-action trajectory. We do not observe significant differences between these two visualizations.

## 5.3 Additional Experiments

To further study the generalization of our observations, we perform four additional experiments.

### 5.3.1 RL Algorithm

To test how our findings generalize to RL algorithms other than PPO, we repeat our experiments by training deep RL policies with the stochastic actor-critic (SAC) [Haarnoja et al., 2018] algorithm. The experimental setup for SAC and its results are documented in detail in Appendix F. Region densities observed early on during training are quite different for SAC than they are for PPO, exhibiting high observed densities which then rapidly drop, before rising again like the PPO case. We hypothesize this is due to a combination of (i) the entropy bonus for SAC, which is then annealed away, and (ii) the different network initialization used for the baseline SAC and PPO implementations. Despite this initial difference, the evolution patterns of densities are consistent with the PPO results later on during

the course of training. Another observation is that similar to PPO results, densities over fixed, current, and random-action trajectories are within the same range. This supports our surprising finding that the complexity of RL policies is not principally captured by increased density on-and-around the optimal trajectories.

### 5.3.2 Environment

We examine whether a non-cyclic task, such as LunarLander, exhibits different linear-region evolution behavior compared to our four default tasks, which are naturally biased towards cyclic locomotion. From the results which are available in Appendix H, we can see that the transition counts and densities along the fixed trajectory are similar in structure to those of the cyclic locomotion tasks, showing that a gradual increase in density appears to be a general property for the PPO setting, even for non-cyclic tasks.

### 5.3.3 Value Network Analysis

To study the difference between the value space and policy space, we repeat our experiments on the value networks trained on HalfCheetah with PPO. Full set of results for this experiment is available in Appendix I. Our results show that linear regions in the value functions largely evolve in a very similar way to those in the policy.

### 5.3.4 Behavior Cloning

Deep RL provides us with a grounded setting to explore the impact of non-IID data on decision regions. To test if the non-IID setting makes a difference, we repeat our experiments by training networks with behavior cloning (BC) using expert data from previously trained policies with PPO on HalfCheetah. The experimental setup for BC and the full results are documented in detail in Appendix J. These results show that the evolution of region densities is different for BC than they are for PPO as the general increase in densities is not observed. In addition, number of transitions, density values, and length of current trajectories are significantly smaller for BC-trained policies. We have two hypotheses for these differences: (i) Policy's learning history plays a significant role in the evolution of regions, as early trajectories inform the template cell divisions that later evolve with further training. Since BC policies observe the entire state space from the moment training starts, they may be able to better divide their state space and avoid adding too much granularity to certain areas. (ii) Network initialization largely affects the resulting linear regions and their evolution during training.

## 6    Conclusions

The structure of ReLU-based reinforcement learning policies remains poorly understood. In this work, we explored a key property of such policies, the density of linear regions observed along policy trajectories, as a proxy for the complexity of functions they can learn. We performed a set of experiments to see how this evolves during training and whether and how changes in depth and width of the policy network affect it. The hypothesis that the final policy trajectory sees an increase in region density is supported to a limited degree, as we observe a moderate increase in density during training, i.e., 50%. This result is consistent across tasks and across policy networks of different configurations. When observed from the vantage point of the evolving current trajectory, the region density is high to begin with, due to the limited length of early trajectories, and then decreases asymptotically towards the observed final density. For the tasks we consider, trajectory lengths are observed to grow and asymptotically approach a plateau during training. Overall, the surprising implication, for our experimental setting, is that the complexity of RL policies appears to be captured by fine-tuning during learning, rather than a significant growth in the complexity of the functions observed along trajectories of the final policy.

The key properties of linear regions as demonstrated for supervised learning of image classifiers, e.g., [Hanin and Rolnick, 2019a], are shown to generally also hold true for reinforcement learning policies. In particular, their expressivity is well modeled as being proportional to the number of neurons, and largely independent of their width and depth, for ranges of these parameters where good policies can still be learned. The number of regions observed for randomly oriented line segments through the origin of the input space is also in line with earlier theoretical and empirical results, and does not

change significantly during training. On a different note, properties of linear regions as measured along policy trajectories observed for policies trained with RL, do not hold for policies trained with BC. This emphasizes the unique behavior of RL compared to its supervised learning counterpart.

Our work comes with a number of limitations. Our experiments are mainly focused on locomotion-based tasks which produce cyclic trajectories. This could mean that the results may not generalize to other types of tasks. For example, some tasks may not result in transition counts that grow, as generally observed in our experiments. In addition, our work studies deep RL policies trained with PPO and SAC. Thus, a broader analysis of more RL algorithms would shed light on the generality of the observations in this work. Since network initialization largely affects the linear regions, it would be ideal to use the same initialization when comparing different RL algorithms. However, RL algorithms are typically sensitive to network initialization, and changes in common practices in their initialization results in instabilities during training. Currently, we compute only the mean region density for a trajectory; it may be useful to further understand the distribution of observed densities across trajectories. We note that it is common practice in RL to apply input (observation) normalization based on a rolling average of observations seen so far [Andrychowicz et al., 2020], and thus our results also employ this strategy. Applying RL without this normalization may impact both the performance of the RL algorithm and the resulting evolution of transition counts and region density.

Our work suggests opportunities for improving our understanding of deep reinforcement learning through studying how states are mapped to actions and how these mappings evolve during training. For instance, a number of questions for future work include understanding the behavior of observed linear regions in the contexts of other RL algorithms, and behavior cloning. Moreover, we envisage various future directions related to this work, including: (i) understanding the impact of non-IID data on decision regions more generally; (ii) consideration of RL algorithms that have less "learning path dependence", e.g., via frequent (re)distillation onto randomly reinitialized policies; (iii) using visited regions to efficiently distill more interpretable non-parametric policies; and (iv) in real-world control settings, using knowledge of decision regions, their sizes, and their orderings to better understand exploration and safety issues. Lastly, we believe that exploring possible connections with the lottery-ticket hypothesis [Frankle and Carbin, 2018] will be worthwhile.

As foundational research related to reinforcement learning, this work could be used to develop improved reinforcement learning policy structures and algorithms, with many possible applications, both good and bad in terms of societal impact.

## Acknowledgments

This work was supported by the NSERC grant RGPIN-2020-05929. David Rolnick acknowledges support from the Canada CIFAR AI Chairs program. This research was enabled in part by technical support and computational resources provided by Compute Canada (`www.computecanada.ca`).

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
