# Appendix

## A  Non-Linearity Definitions

The following activation functions are discussed or used in this work:

1. ReLU [Nair and Hinton, 2010]: $\max(x, 0)$
2. Tanh (Hyperbolic tangent): $(e^x - e^{-x})/(e^x + e^{-x})$
3. Sigmoid: $1/(1 + e^{-x})$

## B  Experimental Setup

All experiments are implemented in Pytorch [Paszke et al., 2019], using the RL codebase from Stable Baselines3 [Raffin et al., 2021]. We conduct our main analysis on four continuous control tasks from the OpenAI gym benchmark suits [Brockman et al., 2016] including HalfCheetah-v2, Walker-v2, Ant-v2, and Swimmer-v2 environments, using PPO algorithm [Schulman et al., 2017].

All policy network configurations are chosen such that the network is fully-capable of learning the task and achieves near state-of-the-art cumulative rewards on the particular task it is trained on. Except where specified, network initialization and hyperparameters are set to the defaults of the PPO implementation of Stable Baselines3. We train our policy networks on 2M samples (i.e. 2M timesteps in the environment). Table 1 shows the range of the return per environment, across all network configurations and random seeds. Return is measured as the mean trajectory cumulative reward across the batch collected at each epoch.

Table 1: Performance Metrics

| 1M Step Return | Average | Max | Min | Std |
|---|---|---|---|---|
| HalfCheetah | 3920.5 | 5035.7 | 2576.1 | 423.2 |
| Walker | 1605.9 | 4527.7 | 135.5 | 1305.6 |
| Ant | 1898.1 | 2609.8 | 1255.9 | 337.2 |
| Swimmer | 82.3 | 128.8 | 30.8 | 28.8 |
| 2M Step Return | Average | Max | Min | Std |
| HalfCheetah | 4876.6 | 6679.4 | 3406.3 | 494.3 |
| Walker | 3618.4 | 6066.1 | 683.3 | 1610 |
| Ant | 2626.6 | 4543.5 | 1535.3 | 672 |
| Swimmer | 112.8 | 132.7 | 41.5 | 19.7 |

## C  Network Dimensions

In all experiments, the policy and value function network architectures used are MLPs with ReLU activations in all hidden layers. Except where specified, the value function has $N = 128$ neurons with layer widths $64, 64$ while the policy network has 18 structures listed in Table 2.

Table 2: Network architectures

| Neurons | | Layer Widths | |
|---|---|---|---|
| N=32 | 32 | 16,16 | 8,8,8,8 |
| N=64 | 64 | 32,32 | 16,16,16,16 |
| N=128 | 128 | 64,64 | 32,32,32,32 |
| N=48 | 48 | 24,24 | 16,16,16 |
| N=96 | 96 | 48,48 | 32,32,32 |
| N=192 | 192 | 96,96 | 64,64,64 |

## D  Initialization and Hyperparameters

The initialization of the policy network has a high impact on the training performance with PPO [Andrychowicz et al., 2020]. To be able to focus on the evolution of linear regions disentangled from properties such as trainability, which is outside the scope of this work, we use the default initialization of PPO from Stable Baselines3. This initialization scheme is consistent with the best-practice recommendations for on-policy RL [Andrychowicz et al., 2020], but it differs from that used in Hanin and Rolnick [2019a]. In this initialization, for both the policy and value function networks, weights of the hidden layers use orthogonal initialization with scaling $\sqrt{2}$, and the biases are set to $0$. The weights of the policy network's output layer are initialized with the scale of $0.01$, and the weights of the value function network's output layer are initialized with the scale of $1$. In Hanin and Rolnick [2019a], networks are initialized with He normalization, and biases are drawn i.i.d from a normal distribution centered at $0$ with variance $10^{-6}$.

We mainly adopt the default hyperparameters of PPO from the implementation of Stable Baselines3 in all of our experiments on PPO. We present the details of the choices of hyperparameters in Table 3.

Table 3: Hyperparameters

| Hyperparameter | Value |
|---|---|
| Training steps | $2 \times 10^6$ |
| Learning rate | $3 \times 10^{-4}$ |
| Number of epochs | 125 |
| Minibatch size | 64 |
| Discount ($\gamma$) | 0.99 |
| GAE parameter ($\lambda$) | 0.95 |
| Init. log stdev. | -1.0 |
| Clipping parameter ($\epsilon$) | 0.2 |
| VF coeff. ($c_1$) | 0.5 |
| Entropy coeff. ($c_2$) | 0.0 |
| Hardware | CPU |

## E  Plots and Error Bars

Each policy network architecture is trained 5 times with 5 random seeds. All metric plots in this work are shown as the mean and standard deviation across these seeds. For the state space visualization of Figure 3, the sampling grid in input space is obtained by sampling a $300 \times 300$ grid with equally distanced points on a window of size $40 \times 200$ where $x \in [-50, 150]$ and $\dot{x} \in [-20, 20]$. Next, we feed each point of the sampling grid into the policy network to compute its activation pattern. Then, we group all points with the same activation pattern into a single linear region uniquely colored according to its activation pattern. Note that the way of encoding a linear region of a point as described in Section 4 guarantees that each region will be uniquely identified by an activation pattern. We create a color map for regions based on their activation patterns. For the state space visualization of Figure 21, we plot regions within a 2D plane that intersects the 17-dimensional input space of HalfCheetah-v2. To obtain the 2D plane, we sample three random points from a trajectory (final trajectory $\tau^*$ or random-action trajectory $\tau^R$) and find the plane that passes through these points. We consider all the points within a square centered at the circumcenter of the three sampled points. The square is scaled such that it is slightly bigger than the circle. In order to avoid missing small regions, which are dominant in higher dimensional spaces, we do not use the subsampling method used for the visualization of the 2D environment. Instead, we compute the exact locations of the boundaries of the regions using our counting method describe in Section 4. Each region is then colored using a randomly sampled color.

## F  Comparison of PPO and SAC

To shed light on the effect of the choice of RL algorithm, we repeat our experiments with the Soft Actor Critic (SAC) algorithm [Haarnoja et al., 2018]. Similar to our main experimental setup for PPO,

we use the Stable-Baselines3 implementation of the SAC algorithm and conduct our experiments on the four continuous control tasks of HalfCheetah-v2, Walker-v2, Ant-v2, and Swimmer-v2 from the OpenAI gym benchmark suits. Again, we run each experiment with 5 different random seeds and the results show the mean and the standard deviation across the random seeds. The policy network architecture used is an MLP with ReLU activations in all hidden layers. To ensure full training capacity, we choose more parameterized networks in this setup. The value function has two hidden layers with 256 neurons in each layer, while the policy network has 12 structures as listed in Table 4. We adopt the network initialization and hyperparameters of SAC from Stable-Baselines3 and train our policy network on 1M samples. Table 5 shows the range of the random return per environment, across all network configurations and random seeds. Return is measured as the mean trajectory return of ten trajectories collected by running the deterministic policy without action noise once every 8000 steps. We present the details of the choices of hyperparameters in Table 6.

Figures 5-12 of this section, show the side-by-side comparison between the results of SAC trained on HalfCheetah and results of PPO trained on HalfCheetah previously shown in Figure 4. We can see that the observed region densities during training are quite different for SAC than they are for PPO, early on in training. We hypothesize that the many transitions (and therefore corresponding high densities) observed early on in training for SAC is due to a combination of (i) the entropy bonus for SAC, which is then annealed away, and (ii) the different network initialization used for the baseline SAC and PPO implementations.

Figures 13-20 of this section, show the same set of plots for all four environments, now grouped by metric type to enable easier comparison between environments. As can be observed the results for normalized region densities on the fixed final trajectory (Figure 13) are broadly consistent across environments (tasks), with Swimmer being an exception.

Table 4: Policy network architectures in SAC experiments

| Neurons | | Layer Widths | |
|---------|-----|---------|-------------|
| N=32    | -   | 16,16   | -           |
| N=64    | -   | 32,32   | -           |
| N=96    | -   | -       | 32,32,32    |
| N=128   | 128 | 64,64   | 32,32,32,32 |
| N=192   | 192 | 96,96   | 64,64,64    |
| N=256   | 256 | 128,128 | 64,64,64    |

Table 5: Performance Metrics of SAC experiments

| 500K Step Return | Average | Max    | Min    | Std    |
|------------------|---------|--------|--------|--------|
| HalfCheetah      | 5752.8  | 8540.4 | 1573.8 | 1307.7 |
| Walker           | 3746.2  | 4865.2 | 1207.2 | 812    |
| Ant              | 1998.6  | 4982.4 | −55    | 1110.9 |
| Swimmer          | 48.2    | 75.1   | −16.4  | 10.1   |

| 1M Step Return   | Average | Max    | Min    | Std    |
|------------------|---------|--------|--------|--------|
| HalfCheetah      | 6572.9  | 9436.1 | 1703.1 | 1540.6 |
| Walker           | 4238.4  | 5131.5 | 1863   | 568.8  |
| Ant              | 2501.6  | 5248.1 | −115.5 | 1116.7 |
| Swimmer          | 51.2    | 97.4   | 39.3   | 8.4    |

Table 6: Hyperparameters of SAC experiments

| Hyperparameter | Value |
|---|---|
| Training steps | $1 \times 10^6$ |
| Learning rate | $3 \times 10^{-4}$ |
| Number of epochs | 125 |
| Batch size | 256 |
| Discount ($\gamma$) | 0.99 |
| Soft update coefficient ($\tau$) | 0.005 |
| Entropy coeff. (initial) | 0.1 |
| Hardware | CPU |

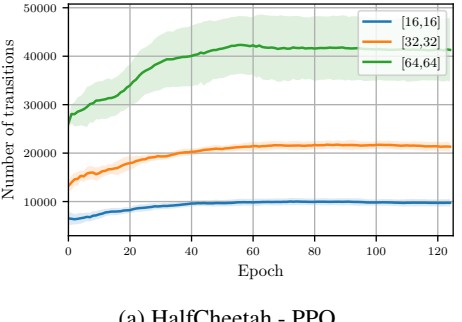

(a) HalfCheetah - PPO

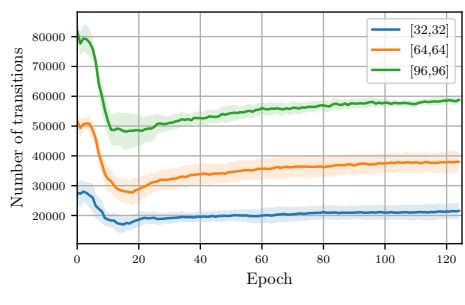

(b) HalfCheetah - SAC

Figure 5: Evolution of the number of transitions over a fixed trajectory sampled from the final fully trained policy ($\tau^*$) during training HalfCheetah with SAC and PPO algorithms. Plots show the mean and standard error across 5 random seeds. In the legend, $[n_1, ..., n_d]$ corresponds to a network architecture with depth $d$ and $n_i$ neurons in each layer.

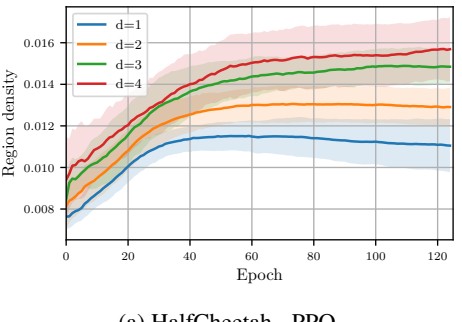

(a) HalfCheetah - PPO

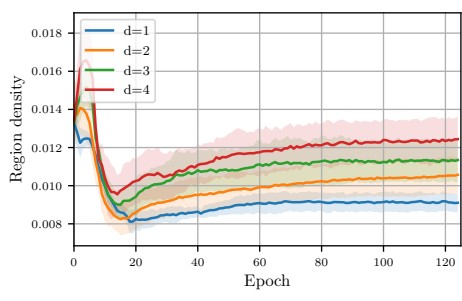

(b) HalfCheetah - SAC

Figure 6: Evolution of the normalized region density over a fixed trajectory sampled from the final fully trained policy ($\tau^*$) during training HalfCheetah with SAC and PPO algorithms.

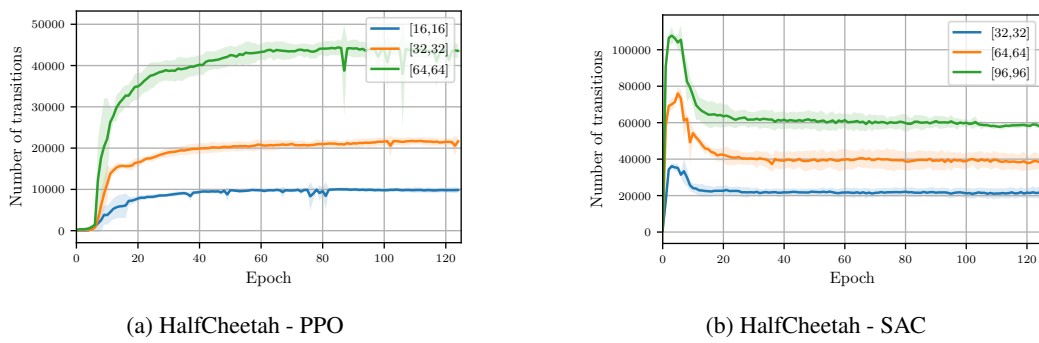

(a) HalfCheetah - PPO

(b) HalfCheetah - SAC

Figure 7: Evolution of the number of transitions over trajectories sampled from the current snapshot of the policy ($\tau$) during training HalfCheetah with SAC and PPO algorithms.

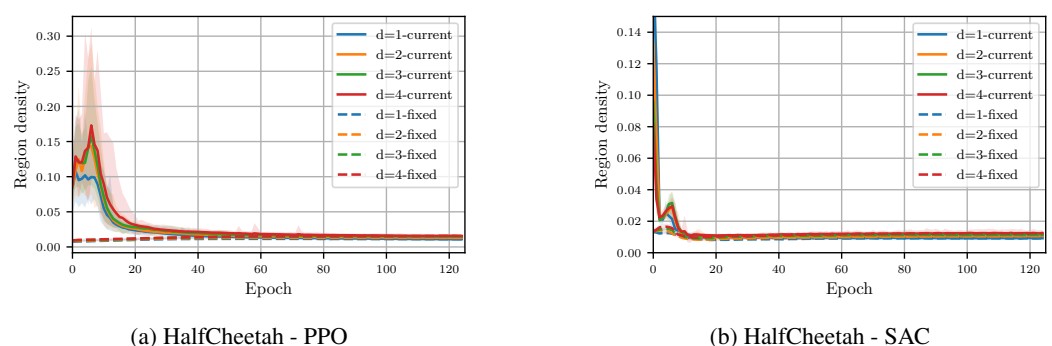

(a) HalfCheetah - PPO

(b) HalfCheetah - SAC

Figure 8: Evolution of the normalized region density over both a fixed trajectory sampled from the final fully trained policy ($\tau^*$) and current trajectories sampled from the current snapshot of the policy ($\tau$) during training HalfCheetah with SAC and PPO algorithms.

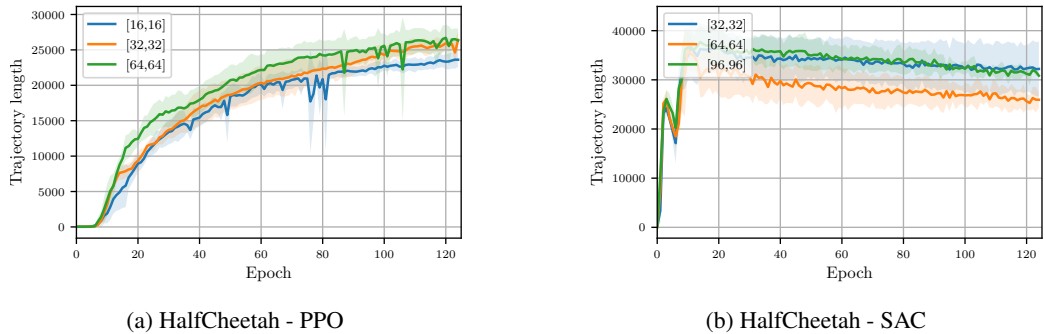

(a) HalfCheetah - PPO

(b) HalfCheetah - SAC

Figure 9: Evolution of the length of trajectories sampled from the current snapshot of the policy ($\tau$) during training HalfCheetah with SAC and PPO algorithms.

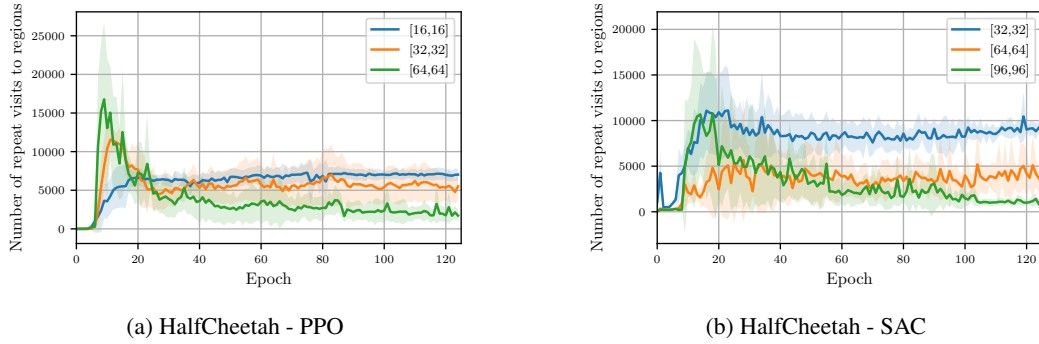

(a) HalfCheetah - PPO

(b) HalfCheetah - SAC

Figure 10: Evolution of the number of repeat visits to regions over trajectories sampled from the current snapshot of the policy ($\tau$) during training HalfCheetah with SAC and PPO algorithms.

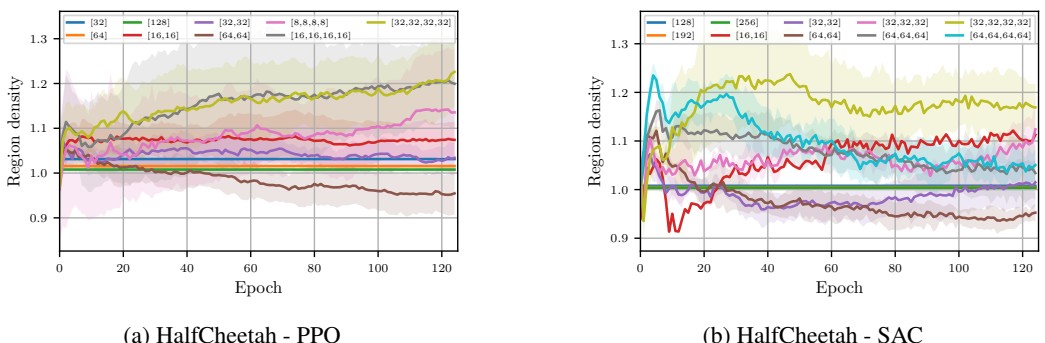

(a) HalfCheetah - PPO

(b) HalfCheetah - SAC

Figure 11: Evolution of the mean normalized region density over 100 random lines passing through the origin during training HalfCheetah with SAC and PPO algorithms. For each policy network configuration, we sample 100 random lines and compute the density of transitions as we sweep along these lines. We then report the mean density observed over these 100 lines.

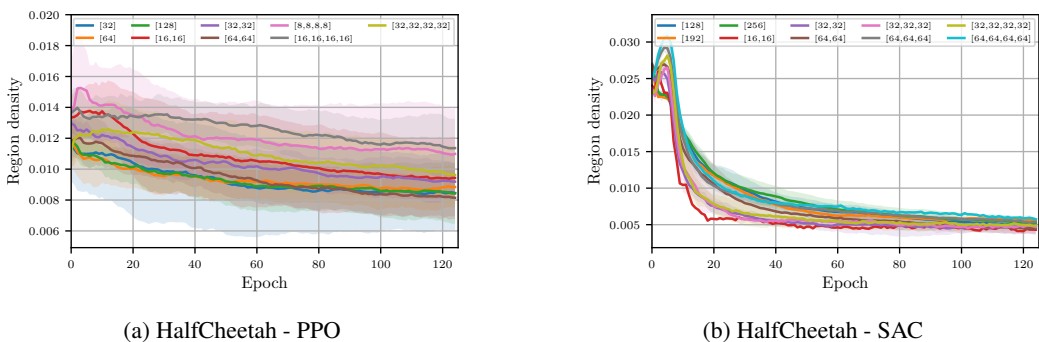

(a) HalfCheetah - PPO

(b) HalfCheetah - SAC

Figure 12: Evolution of the mean normalized region density over 10 random-action trajectories ($\tau^R$) during training HalfCheetah with SAC and PPO algorithms. For each policy network configuration, we sample 10 random-action trajectories and compute the density of transitions as we sweep along these trajectories, and report the mean value of these trajectories.

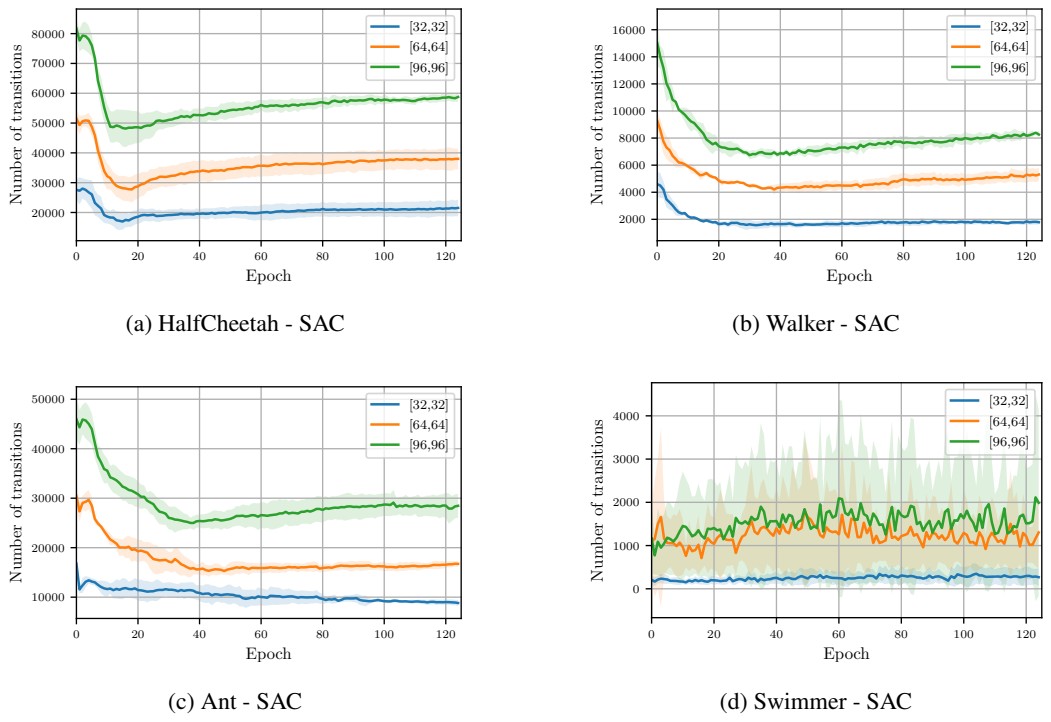

(a) HalfCheetah - SAC

(b) Walker - SAC

(c) Ant - SAC

(d) Swimmer - SAC

Figure 13: Evolution of the number of transitions over a fixed trajectory sampled from the final fully trained policy ($\tau^*$) during training different tasks with SAC.

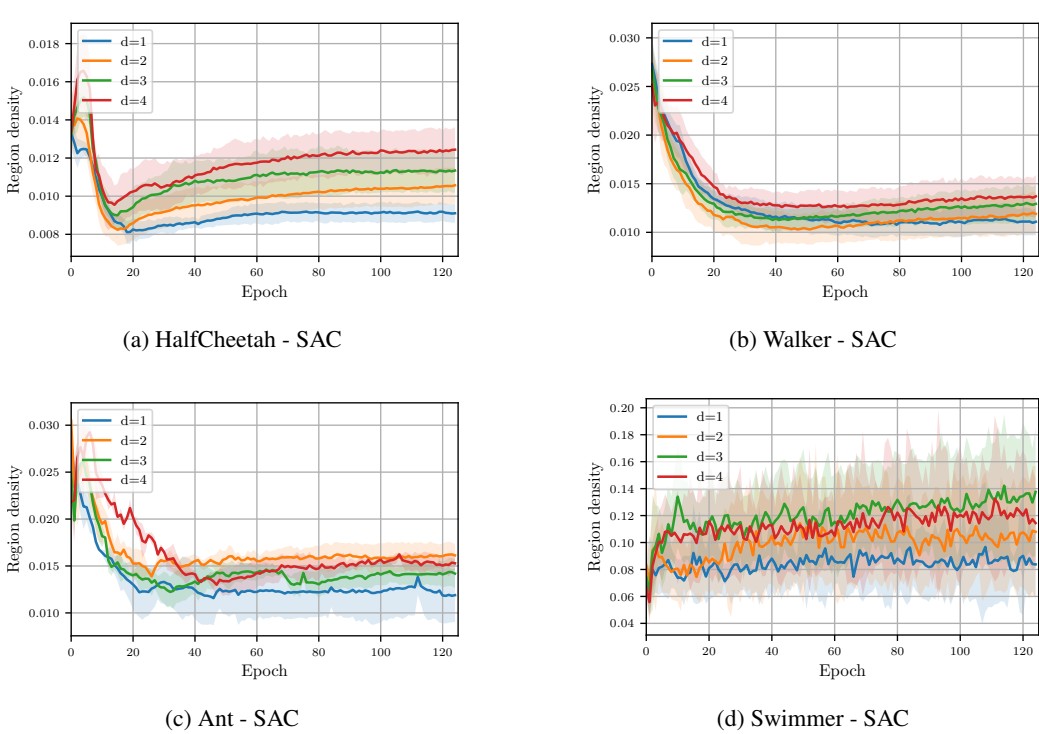

(a) HalfCheetah - SAC

(b) Walker - SAC

(c) Ant - SAC

(d) Swimmer - SAC

Figure 14: Evolution of the normalized region density over a fixed trajectory sampled from the final fully trained policy ($\tau^*$) during training different tasks with SAC.

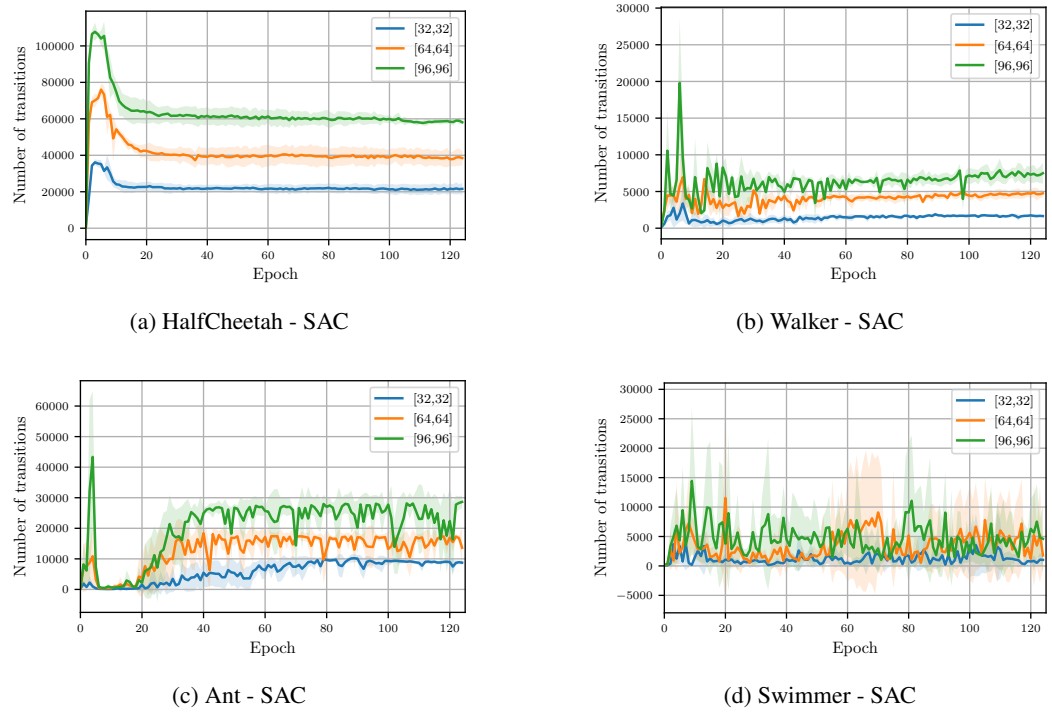

Figure 15: Evolution of the number of transitions over trajectories sampled from the current snapshot of the policy ($\tau$) during training different tasks with SAC.

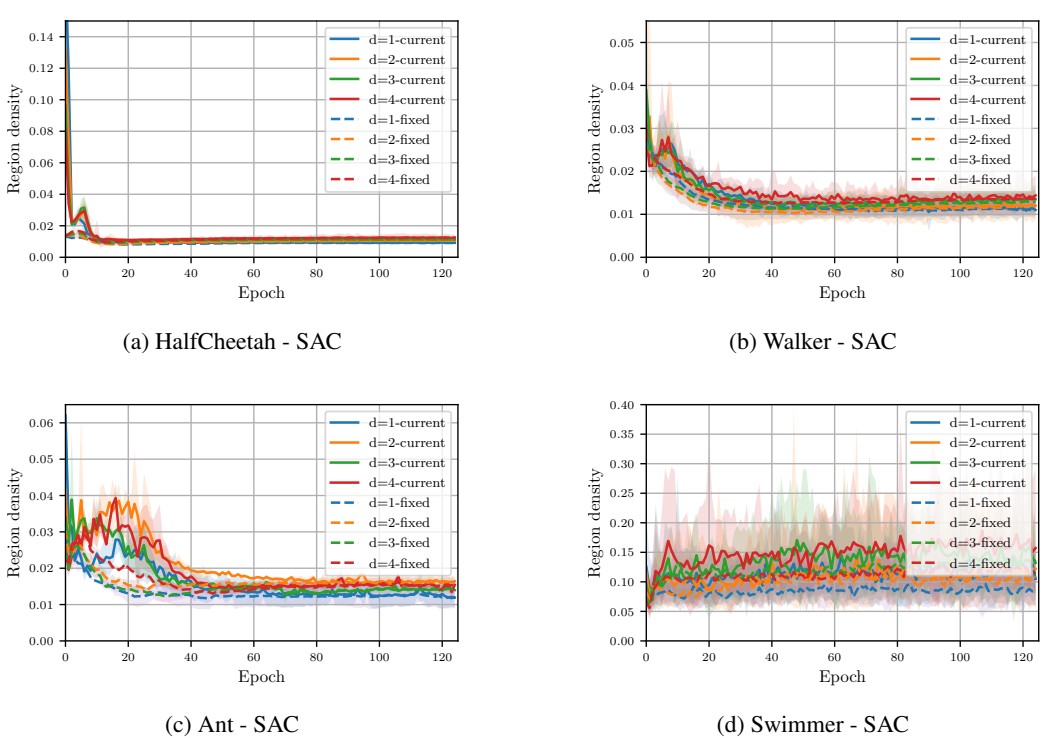

Figure 16: Evolution of the normalized region density over both a fixed trajectory sampled from the final fully trained policy ($\tau^*$) and current trajectories sampled from the current snapshot of the policy ($\tau$) during training different tasks with SAC.

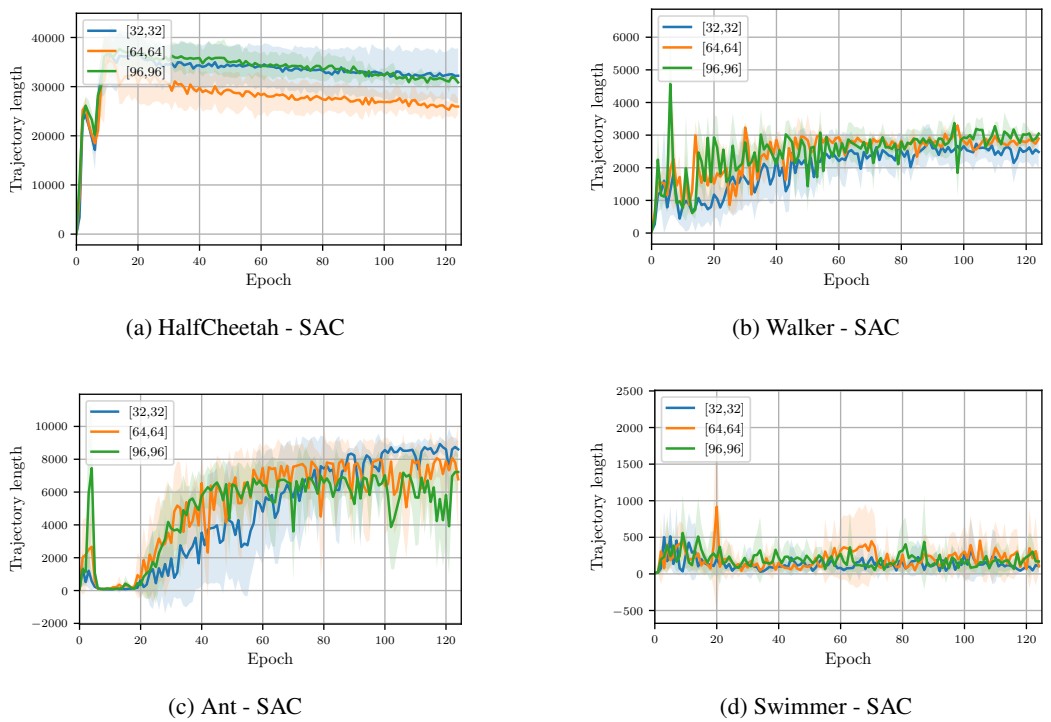

(a) HalfCheetah - SAC

(b) Walker - SAC

(c) Ant - SAC

(d) Swimmer - SAC

Figure 17: Evolution of the length of trajectories sampled from the current snapshot of the policy ($\tau$) during training different tasks with SAC.

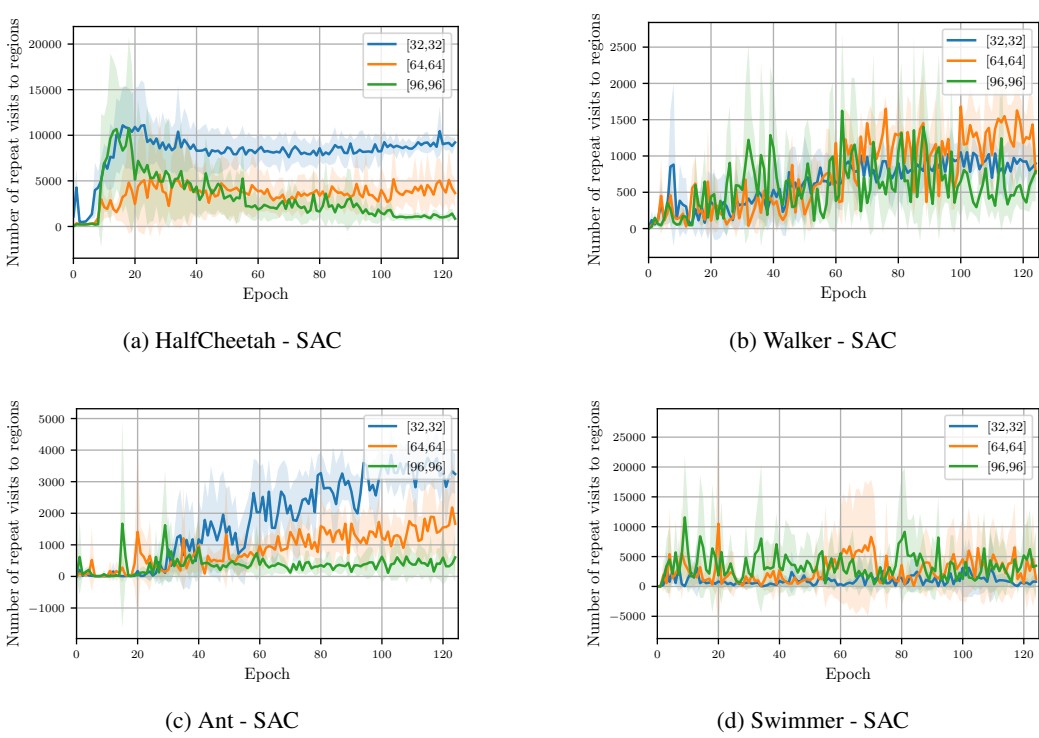

(a) HalfCheetah - SAC

(b) Walker - SAC

(c) Ant - SAC

(d) Swimmer - SAC

Figure 18: Evolution of the number of repeat visits to regions over trajectories sampled from the current snapshot of the policy ($\tau$) during training different tasks with SAC.

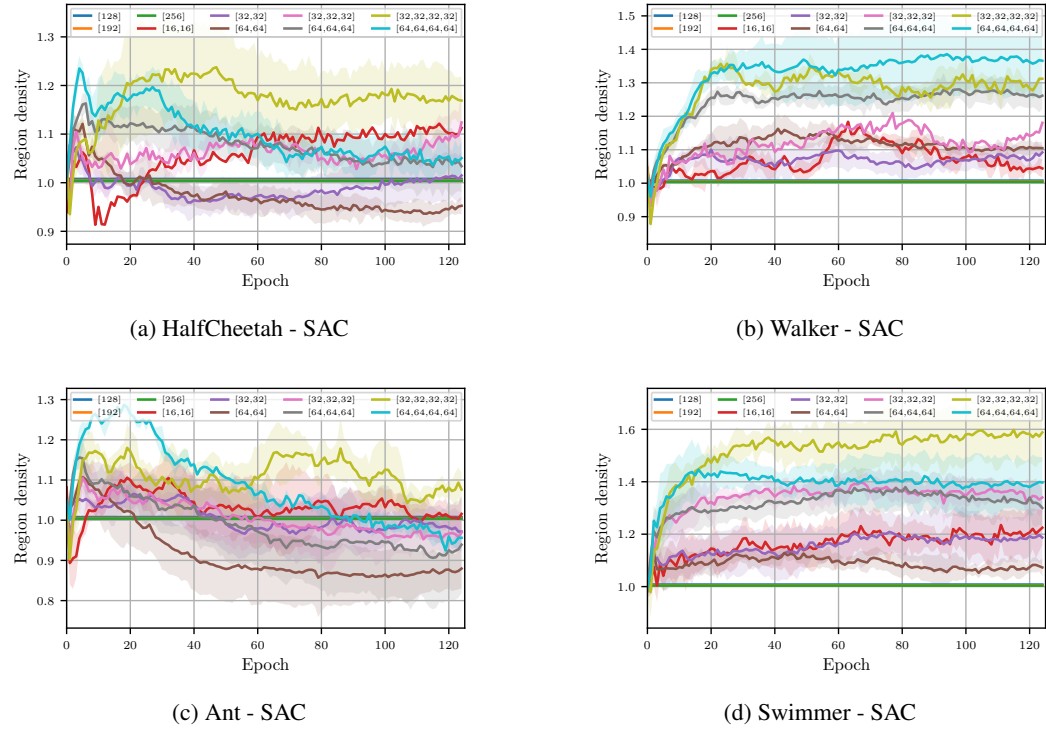

(a) HalfCheetah - SAC

(b) Walker - SAC

(c) Ant - SAC

(d) Swimmer - SAC

Figure 19: Evolution of the mean normalized region density over 100 random lines passing through the origin during training different tasks with SAC. For each policy network configuration, we sample 100 random lines and compute the density of transitions as we sweep along these lines. We then report the mean density observed over these 100 lines.

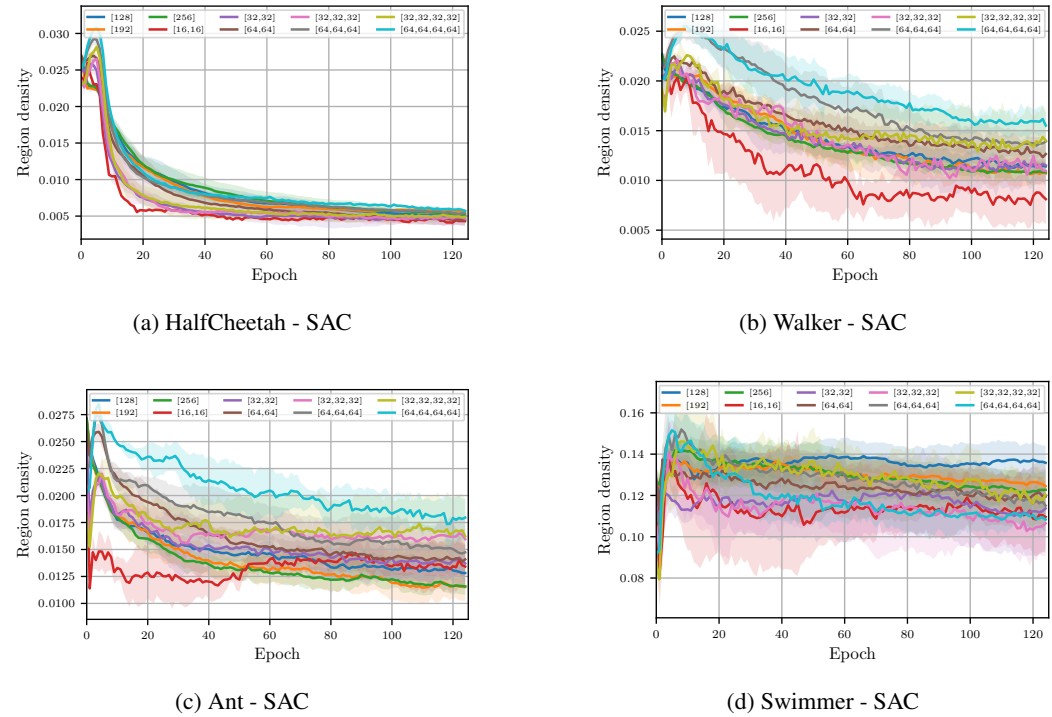

(a) HalfCheetah - SAC

(b) Walker - SAC

(c) Ant - SAC

(d) Swimmer - SAC

Figure 20: Evolution of the mean normalized region density over 10 random-action trajectories $(\tau^R)$ during training different tasks with SAC. For each policy network configuration, we sample 10 random-action trajectories and compute the density of transitions as we sweep along these trajectories, and report the mean value of these trajectories.

## G  Decision Regions viewed via Embedding Planes

To gain insights on what the linear regions may look like in our high dimensional setting, similar to Novak et al. [2018] and Hanin and Rolnick [2019a], we visualize the linear regions over a 2-dimensional slice through the input space defined by three points sampled from a trajectory emerged from the policy. Figure 21 shows the linear regions of a $(32, 32)$ policy network trained on HafCheetah in the 17-dimensional input space, over a 2-dimensional plane crossing three projection points sampled from the final trajectory $(\tau^*)$ in the first row, and sampled from the random-action trajectory $(\tau^R)$ in the second row. Comparing the visualizations of projected on points from $\tau^*$ and $\tau^R$ (first and second row), we do not observe a significant difference between the granularity of regions over points of these two trajectories. Moreover, surprisingly, our visualizations are more consistent with the findings of Novak et al. [2018] with the projection points lying in regions of lower density.

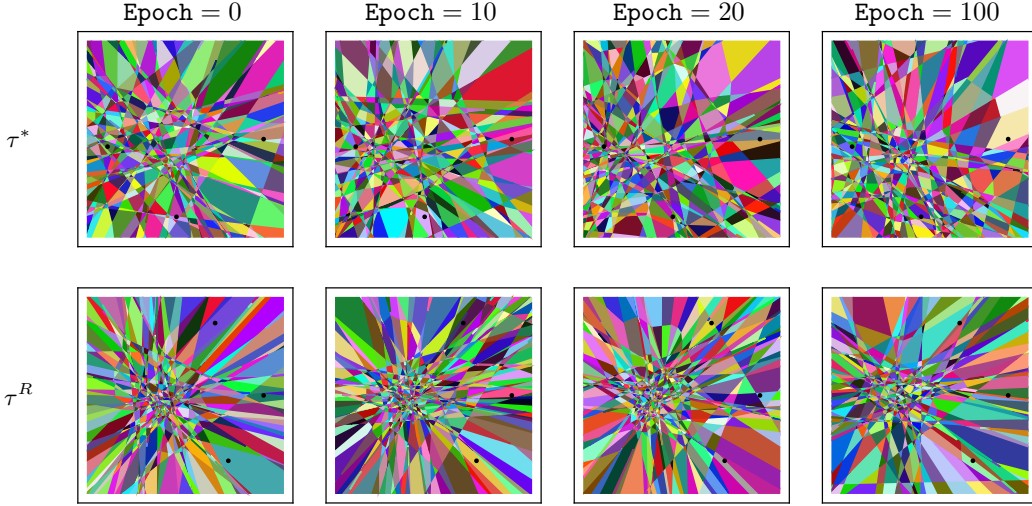

Figure 21: Linear regions that intersect a 2D plane through the input space for a network of depth 2 and width 32 trained on HalfCheetah-v2. Black dots indicate three points from the input space on which the plane is defined. In the first row, the three points are randomly sampled from the final trajectory $\tau^*$ whereas in the second row, the three points are randomly sampled from a random-action trajectory $\tau^R$.

## H  Decision Regions for a Non-Cyclic Task: LunarLander

In this section, we aim to shed light on the possible effects of cyclic nature of locomotion tasks on our previous PPO results. Therefore, we repeat our experiments by training PPO on LunarLanderContinuous-v2 from OpenAI gym. Figure 22 shows the plots from this experiment. The linear-region evolution behavior is in fact still similar with the only discrepancy being the evolution pattern of observed density over current trajectories. For LunarLander, observed density over current trajectories does not decrease during training while having the same range of values as the density over fixed and random-action trajectories. This is due to the difference between the nature of non-cyclic and cyclic tasks. For locomotion tasks, the length of the cyclic trajectories increases during training, while for LunarLander, the trajectory length initially increases, then plateaus once the agent converges.

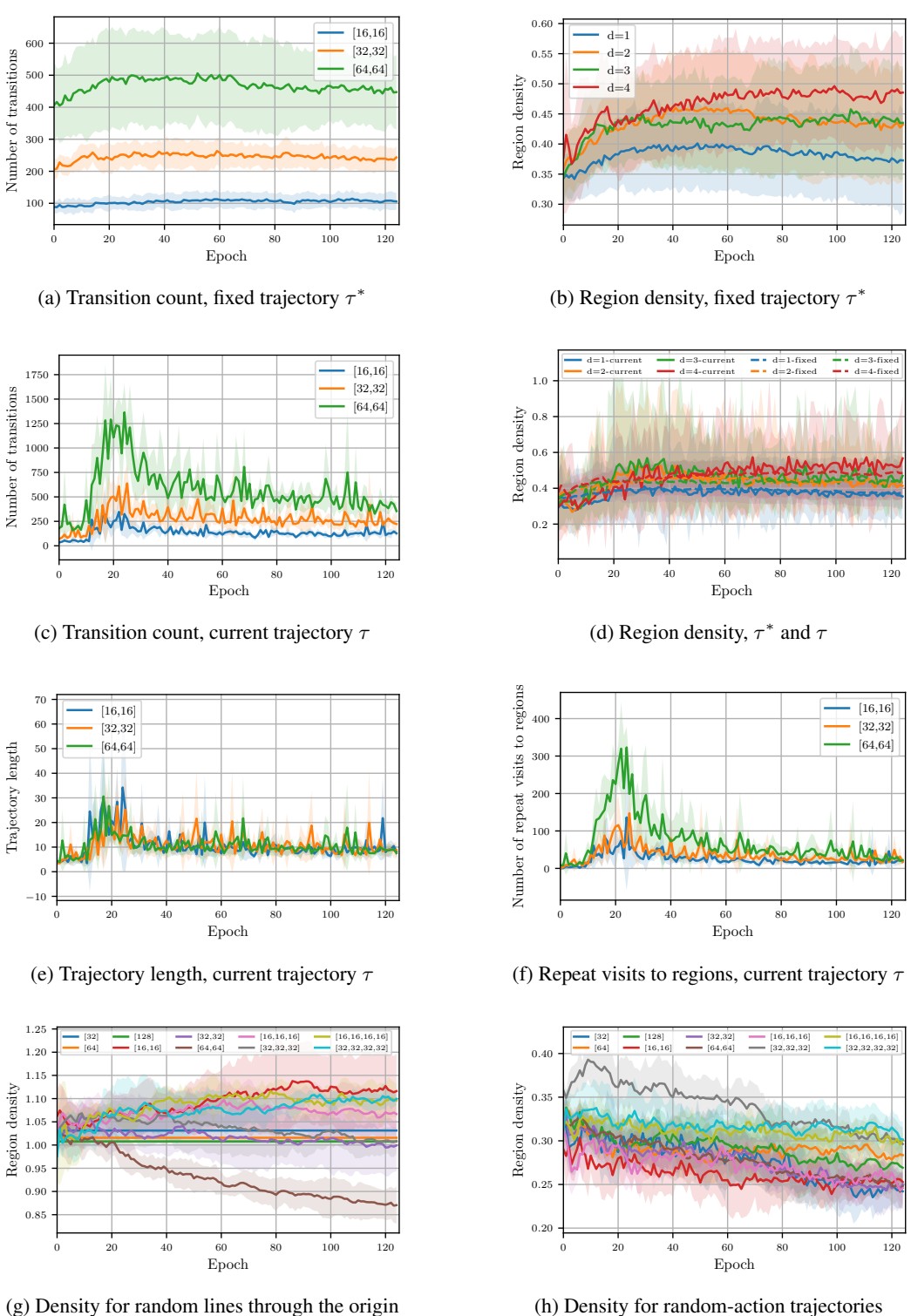

(a) Transition count, fixed trajectory $\tau^*$

(b) Region density, fixed trajectory $\tau^*$

(c) Transition count, current trajectory $\tau$

(d) Region density, $\tau^*$ and $\tau$

(e) Trajectory length, current trajectory $\tau$

(f) Repeat visits to regions, current trajectory $\tau$

(g) Density for random lines through the origin

(h) Density for random-action trajectories

Figure 22: LunarLander Results. Training is with PPO and the ranges indicate the standard error across 5 random seeds. In the legend $[n_1, ..., n_d]$ describes a network architecture of depth $d$ and $n_i$ neurons in each layer. Summary results are grouped by network depth, $d$.

# I  Value Network Analysis

We also explore decision regions defined by the value-function network. We repeat our main experiments on the value network of PPO agents. More particularly, we train a policy network with a fixed network structure of $(64, 64)$ with 18 different value network configurations on HalfCheetah using the PPO algorithm. The value network configurations used here are the same as the ones used for the policy network and are listed in Table 2, having $N \in \{32, 48, 64, 96, 128, 192\}$ neurons, widths $w \in \{8, 16, 32, 64\}$, and depths $d \in \{1, 2, 3, 4\}$. These network configurations are chosen such that the network is fully-capable of learning the task and achieves near state-of-the-art cumulative reward on the particular task it is trained on. To compute the metrics, we now use the weights and biases of the value function network, instead of the policy network. Figure 23 shows the resulting plots. In comparing Figure 23 and Figure 4, we see that the evolution plots of the region densities are remarkably similar in structure, for both fixed and current trajectories.

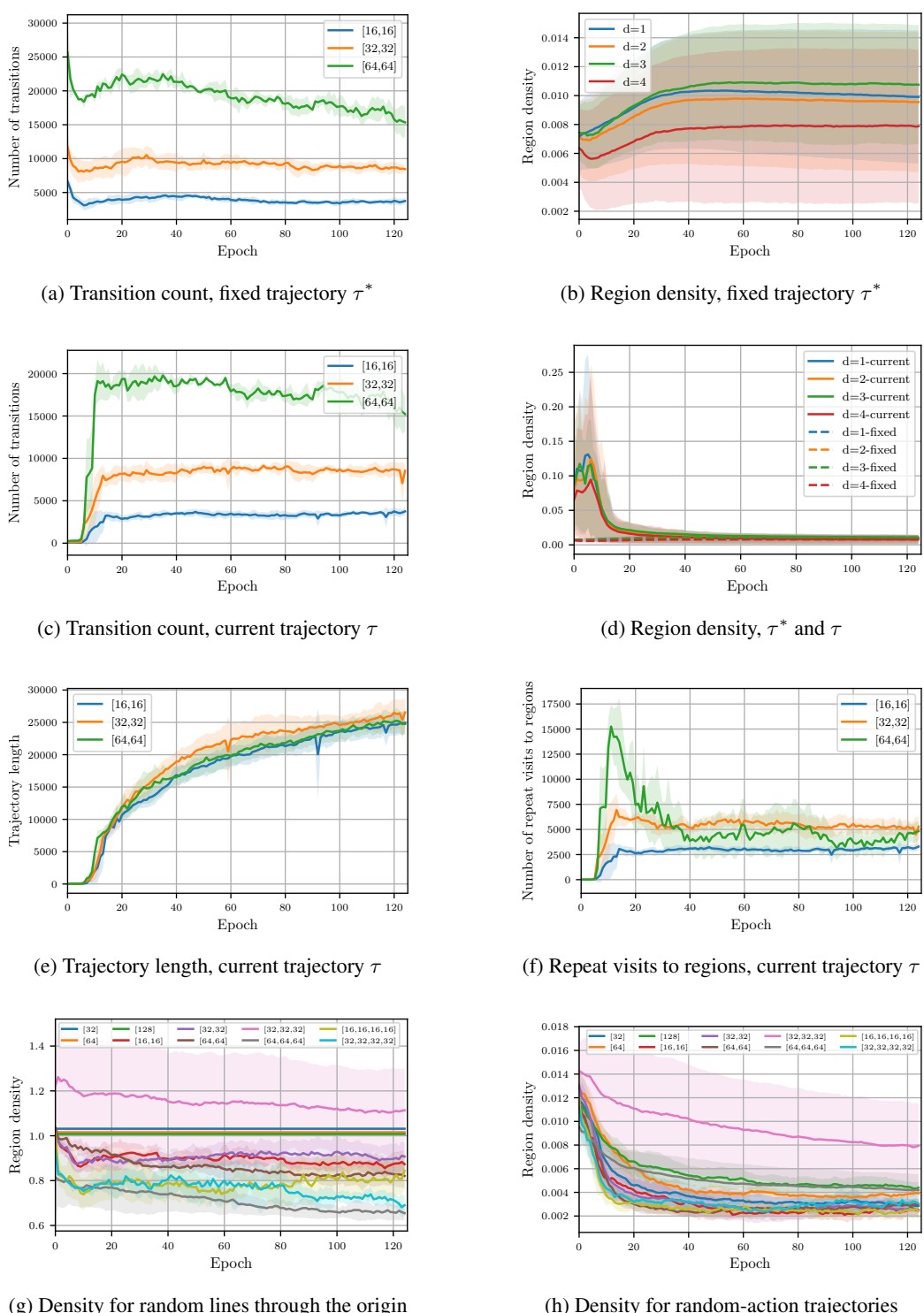

(a) Transition count, fixed trajectory $\tau^*$

(b) Region density, fixed trajectory $\tau^*$

(c) Transition count, current trajectory $\tau$

(d) Region density, $\tau^*$ and $\tau$

(e) Trajectory length, current trajectory $\tau$

(f) Repeat visits to regions, current trajectory $\tau$

(g) Density for random lines through the origin

(h) Density for random-action trajectories

Figure 23: Evolution of the number of transitions, linear region densities, and length of trajectories for the value function, during training HalfCheetah with PPO. The ranges indicate the standard error across 5 random seeds. $[n_1, ..., n_d]$ in the legend corresponds to a network architecture with depth $d$ and $n_i$ neurons in each layer. Summary results are grouped by network depth, $d$.

# J   Behavior Cloning

We perform an additional experiment to study whether linear regions emerging from policies trained with deep RL on non-IID data are different from those emerging from policies trained with BC on IID data. For a direct comparison, we repeat our experiments by first collecting expert data from each of the 18 network architectures previously trained with PPO on HalfCheetah. For each expert PPO policy, we initialize a neural network policy with the same architecture and train it using BC on the collected expert data. We then evaluate this new set of trained policies using the same evaluation method.

For policy architecture $\mathcal{A}$ previously trained with PPO on HalfCheetah, we randomly pick one of the 5 experts (from the random seeds), and collect 500 episodes of expert data by letting the policy interact with the environment. Actions are sampled from the stochastic policy. We then train a new policy network with the same architect $\mathcal{A}$ using BC on the collected dataset. We repeat each experiment with 5 random seeds. To ensure BC-trained policies are fully trained, we choose hyperparameters such that the policy achieves similar average episodic return to its expert PPO-trained policy. We present the details of the choices of hyperparameters in Table 7. We use the default initialization of linear layers used by PyTorch where weights are He normal with gain=$\sqrt{5}$, and biases are IID normal with variance $1/\texttt{fan-in}$.

Table 7: Hyperparameters of BC experiments

| Hyperparameter | Value |
| --- | --- |
| Training epochs | 125 |
| Learning rate | $10^{-3}$ |
| Batch size | 64 |
| Optimizer | SGD |
| Hardware | CPU |

Figures 24-31 of this section, show the side-by-side comparison between the results of BC trained on HalfCheetah and results of PPO trained on HalfCheetah previously shown in Figure 4. We can see that the observed region densities during training are much smaller for BC than they are for PPO and that the general trend of increased density is not visible for BC. We hypothesize that these differences are due to a combination of (i) different evolution of linear regions for networks trained with RL and supervised learning due to the inherent difference in levels of information about the state space available to each methodx (ii) the different network initializations used for the BC and PPO implementations.

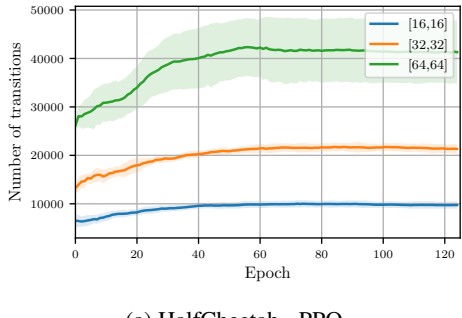
(a) HalfCheetah - PPO

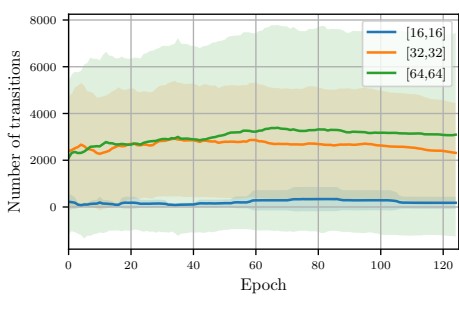
(b) HalfCheetah - BC

Figure 24: Evolution of the number of transitions over a fixed trajectory sampled from the final fully trained policy ($\tau^*$) during training HalfCheetah with BC and PPO algorithms. Plots show the mean and standard error across 5 random seeds. In the legend, $[n_1, ..., n_d]$ corresponds to a network architecture with depth $d$ and $n_i$ neurons in each layer.

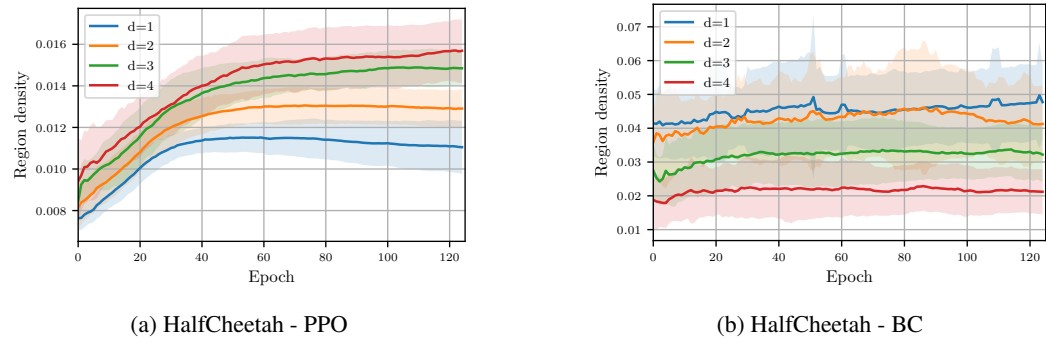

(a) HalfCheetah - PPO

(b) HalfCheetah - BC

Figure 25: Evolution of the normalized region density over a fixed trajectory sampled from the final fully trained policy ($\tau^*$) during training HalfCheetah with BC and PPO algorithms.

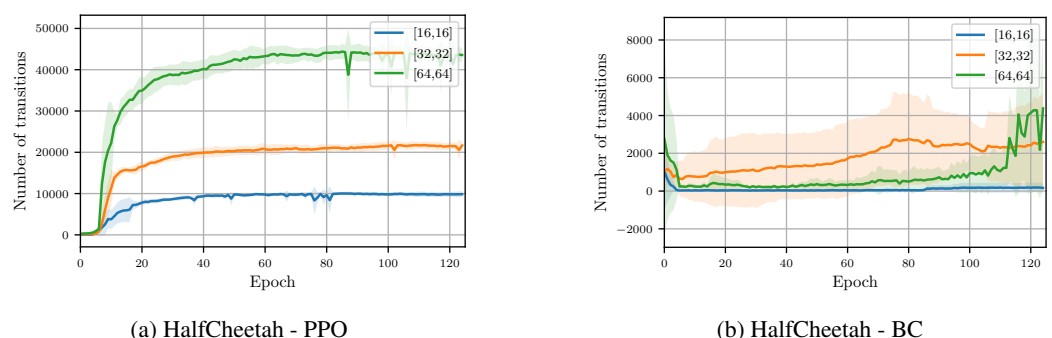

(a) HalfCheetah - PPO

(b) HalfCheetah - BC

Figure 26: Evolution of the number of transitions over trajectories sampled from the current snapshot of the policy ($\tau$) during training HalfCheetah with BC and PPO algorithms.

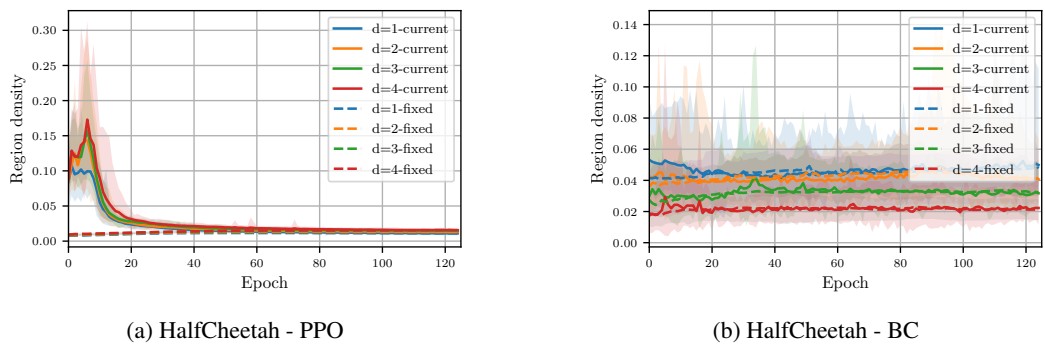

(a) HalfCheetah - PPO

(b) HalfCheetah - BC

Figure 27: Evolution of the normalized region density over both a fixed trajectory sampled from the final fully trained policy ($\tau^*$) and current trajectories sampled from the current snapshot of the policy ($\tau$) during training HalfCheetah with BC and PPO algorithms.

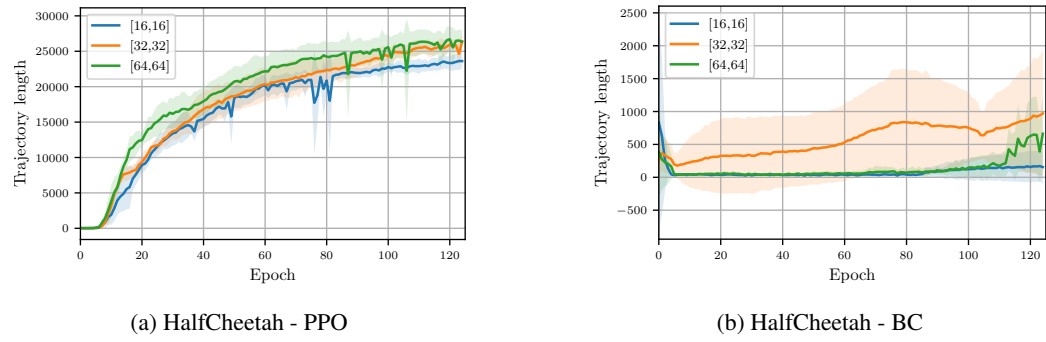

(a) HalfCheetah - PPO

(b) HalfCheetah - BC

Figure 28: Evolution of the length of trajectories sampled from the current snapshot of the policy ($\tau$) during training HalfCheetah with BC and PPO algorithms.

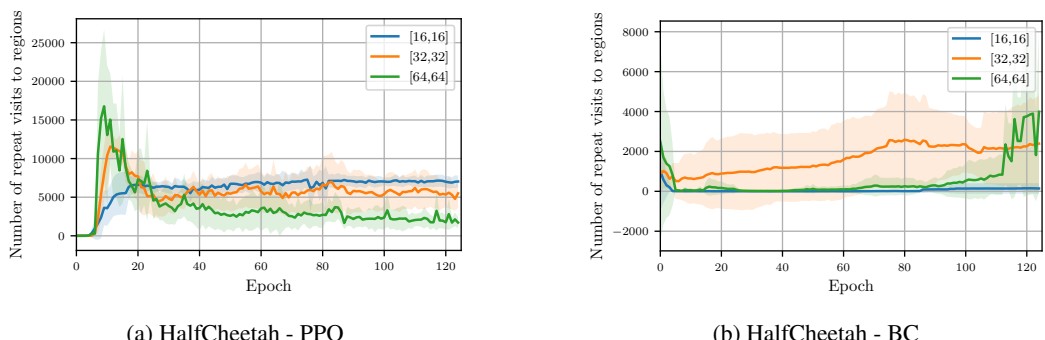

(a) HalfCheetah - PPO

(b) HalfCheetah - BC

Figure 29: Evolution of the number of repeat visits to regions over trajectories sampled from the current snapshot of the policy ($\tau$) during training HalfCheetah with BC and PPO algorithms.

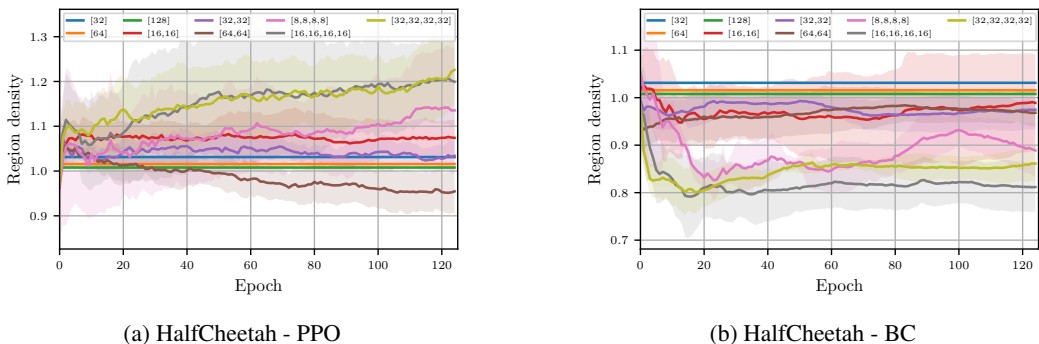

(a) HalfCheetah - PPO

(b) HalfCheetah - BC

Figure 30: Evolution of the mean normalized region density over 100 random lines passing through the origin during training HalfCheetah with BC and PPO algorithms. For each policy network configuration, we sample 100 random lines and compute the density of transitions as we sweep along these lines. We then report the mean density observed over these 100 lines.

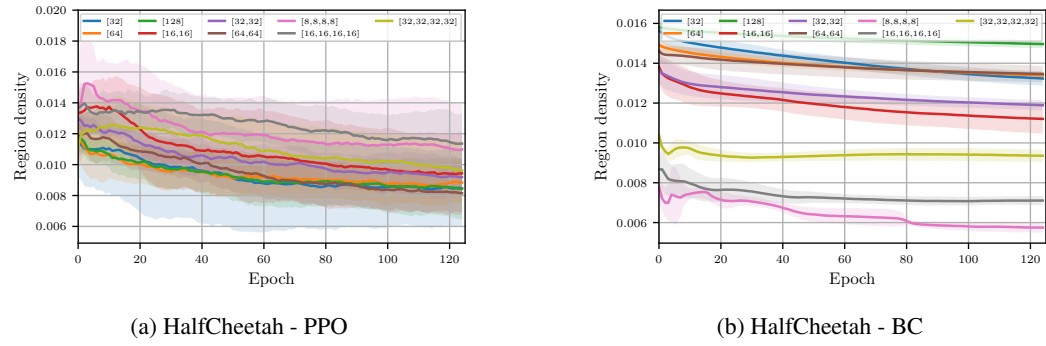

(a) HalfCheetah - PPO

(b) HalfCheetah - BC

Figure 31: Evolution of the mean normalized region density over 10 random-action trajectories ($\tau^R$) during training HalfCheetah with BC and PPO algorithms. For each policy network configuration, we sample 10 random-action trajectories and compute the density of transitions as we sweep along these trajectories, and report the mean value of these trajectories.

# K    Additional Results

We previously provided the results for the HalfCheetah environment in Figure 4. Figures 32-39 of this section, show the same set of plots for the three remaining environments trained with PPO. Plots are now grouped by metric type instead of environment, to enable easier comparison between environments. Figure 40 shows the full set of results of Figures 4(a) and 4(c) for all policy network architectures trained on HalfCheetah.

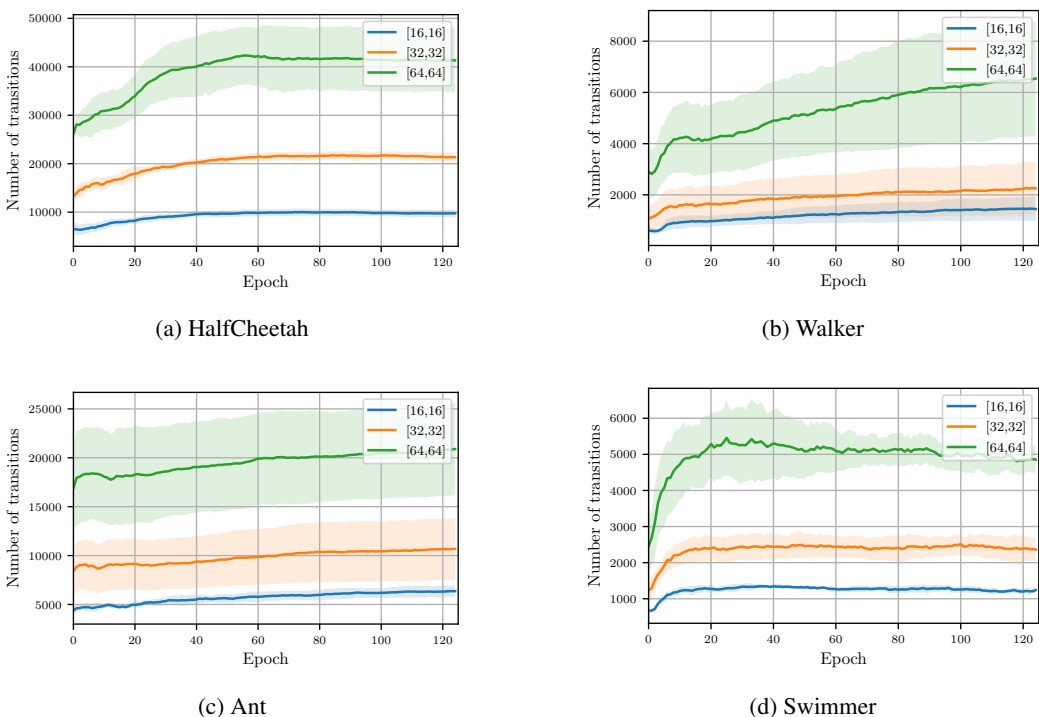

(a) HalfCheetah

(b) Walker

(c) Ant

(d) Swimmer

Figure 32: Evolution of the number of transitions over a fixed trajectory sampled from the final fully trained policy ($\tau^*$) during training for different tasks. Plots show the mean and standard error across 5 random seeds. In the legend, $[n_1, ..., n_d]$ corresponds to a network architecture with depth $d$ and $n_i$ neurons in each layer. These plots show a moderate and gradual increase in the number of transitions over a fixed trajectory observed during training, with larger policy networks having more transitions.

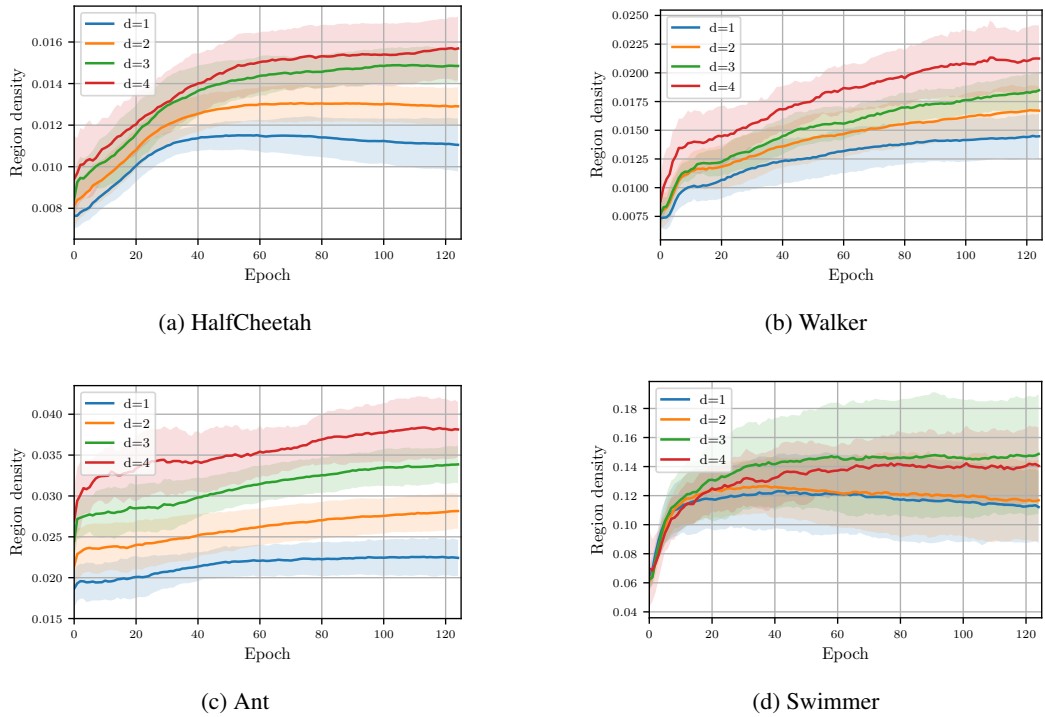

(a) HalfCheetah

(b) Walker

(c) Ant

(d) Swimmer

Figure 33: Evolution of the normalized region density over a fixed trajectory sampled from the final fully trained policy ($\tau^*$) during training for different tasks. Plots show the mean and the standard error over all networks with equal depth. Note that the vertical axes do not begin at zero. These plots show a moderate and gradual increase in the density of transitions over a fixed trajectory during training. We can also observe that deeper policy networks results in moderately denser regions in the learned policies.

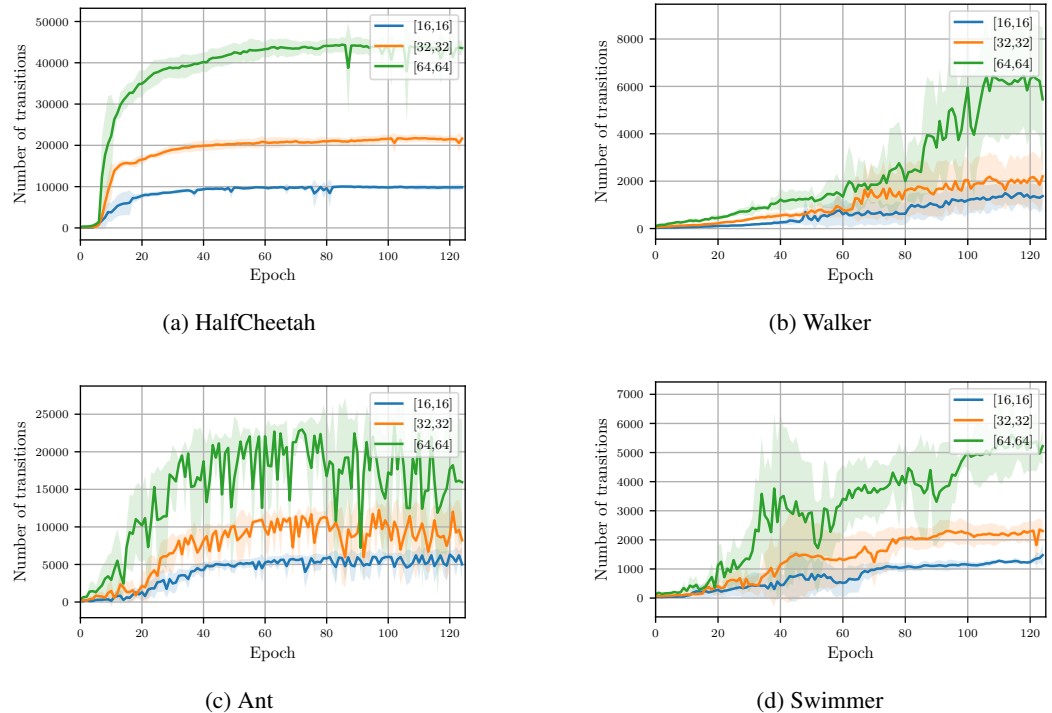

(a) HalfCheetah

(b) Walker

(c) Ant

(d) Swimmer

Figure 34: Evolution of the number of transitions over trajectories sampled from the current snapshot of the policy ($\tau$) during training for different tasks. These plots show an increase in the number of transitions observed on trajectories sampled from current snapshots of the policy during training.

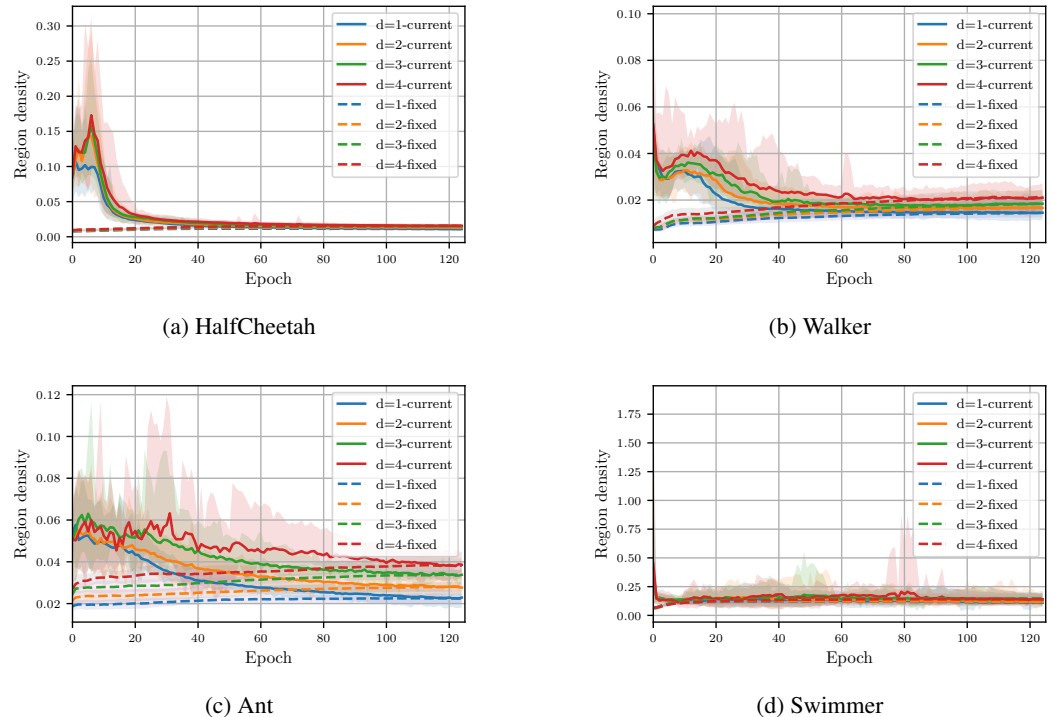

(a) HalfCheetah

(b) Walker

(c) Ant

(d) Swimmer

Figure 35: Evolution of the normalized region density over both a fixed trajectory sampled from the final fully trained policy ($\tau^*$) and current trajectories sampled from the current snapshot of the policy ($\tau$) during training for different tasks. These plots show that the density over fixed and current trajectories converges to the same values over training, while it increases for former and decreases for the latter. We speculate that the density is higher for current trajectories earlier during training due to early exploration and the form of network initialization.

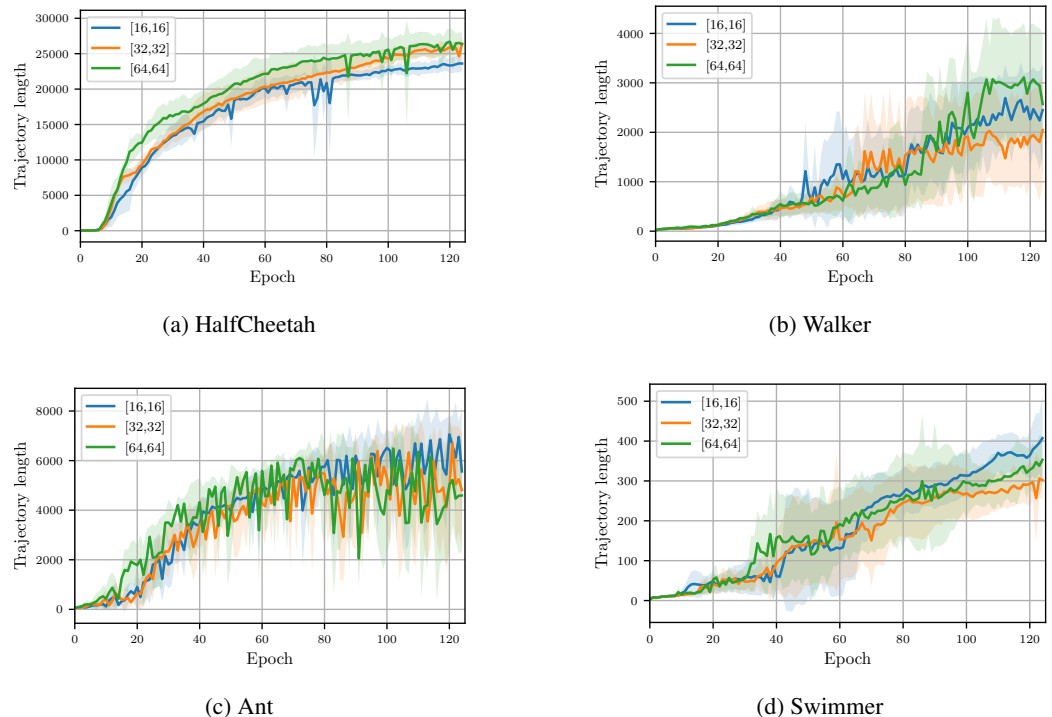

(a) HalfCheetah

(b) Walker

(c) Ant

(d) Swimmer

Figure 36: Evolution of the length of trajectories sampled from the current snapshot of the policy ($\tau$) during training for different tasks. These plots show that the length of the trajectories increase with training.

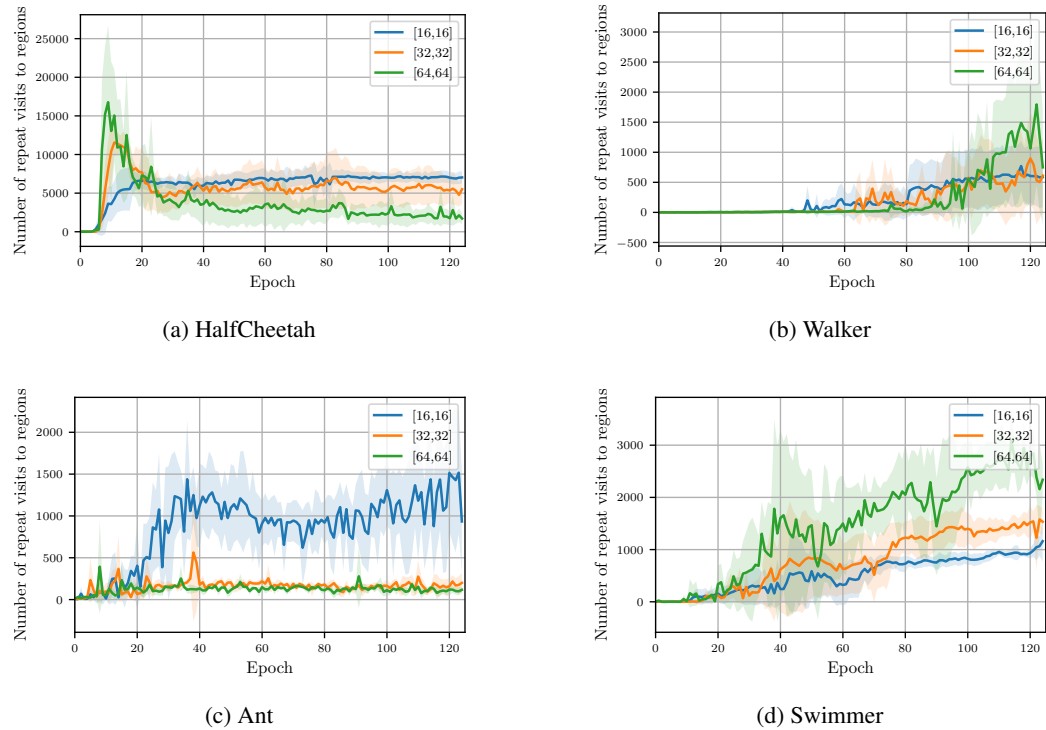

(a) HalfCheetah

(b) Walker

(c) Ant

(d) Swimmer

Figure 37: Evolution of the number of repeat visits to regions over trajectories sampled from the current snapshot of the policy ($\tau$) during training for different tasks. We can see that the number of repeat visits generally increase with training because of the cyclic trajectories resulting from locomotion-based tasks. We speculate that for HalfCheetah, repeat visits are high earlier during training because of limited exploration.

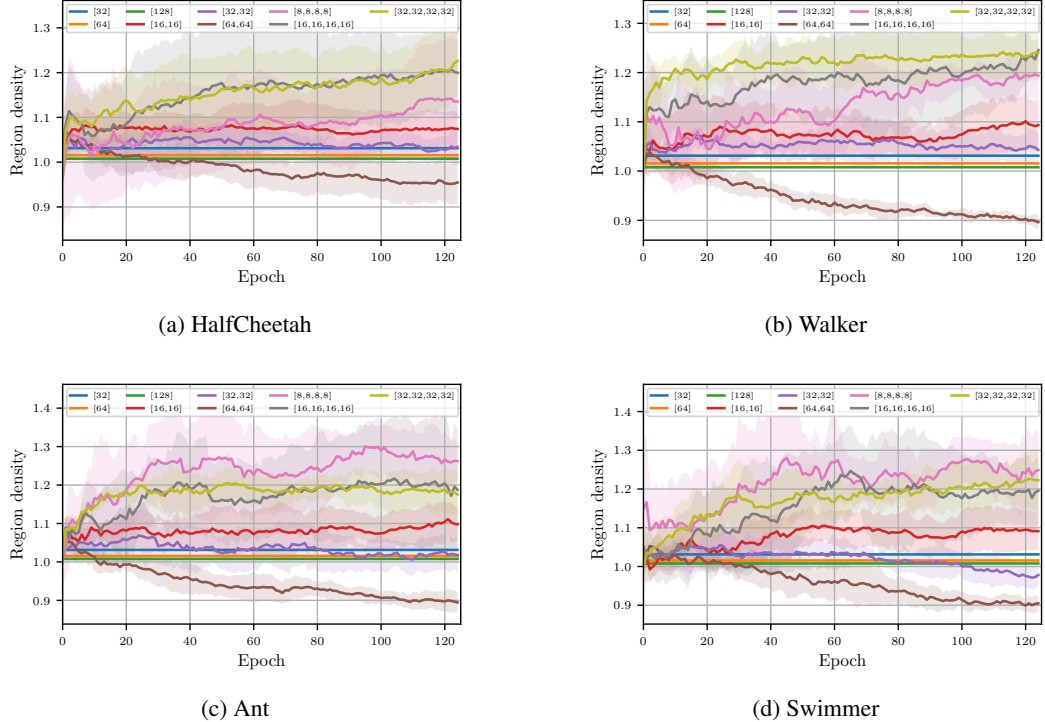

(a) HalfCheetah

(b) Walker

(c) Ant

(d) Swimmer

Figure 38: Evolution of the mean normalized region density over 100 random lines passing through the origin during training for different tasks. For each policy network configuration, we sample 100 random lines and compute the density of transitions as we sweep along these lines. We then report the mean density observed over these 100 lines. These plots show that the mean normalized density starts close to 1 at initialization and remains roughly constant during training which is consistent with the findings of Hanin and Rolnick [2019a]. Note that because the vertical axis does not begin at zero, the variations around 1.0 is scaled.

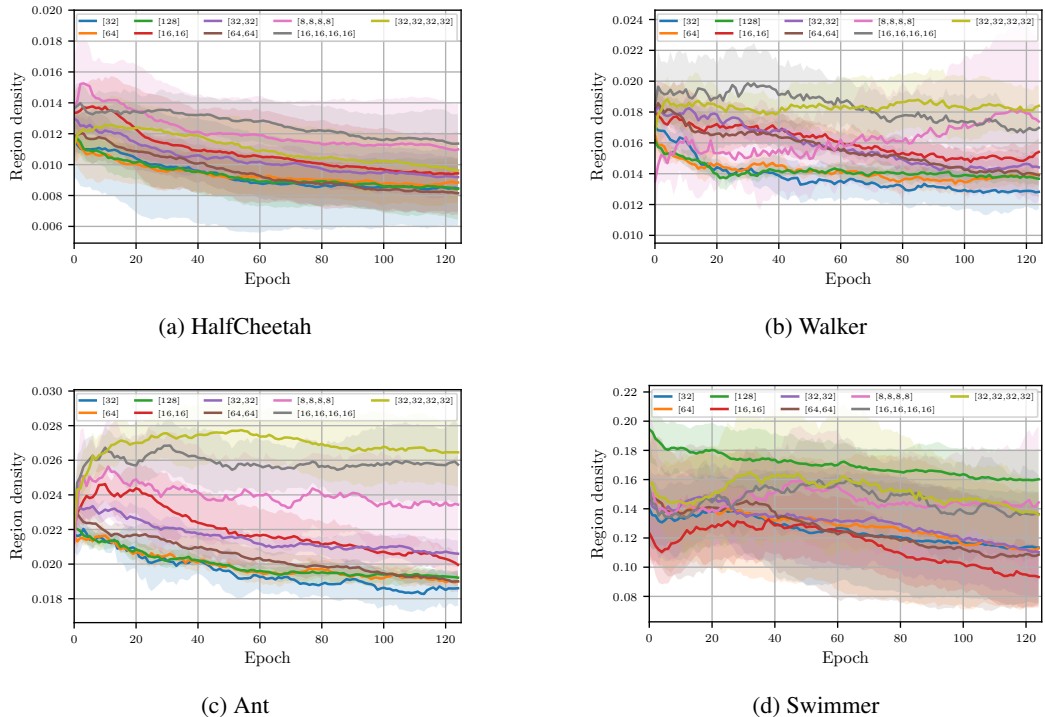

(a) HalfCheetah

(b) Walker

(c) Ant

(d) Swimmer

Figure 39: Evolution of the mean normalized region density over 10 random-action trajectories ($\tau^R$) during training for different tasks. For each policy network configuration, we sample 10 random-action trajectories and compute the density of transitions as we sweep along these trajectories, and report the mean value of these trajectories. These plots show that the observed normalized density for random-action trajectories decreases slightly with training. When compared with the results of Figure 35, we observe that the observed densities are marginally less than that of fixed and current trajectories.

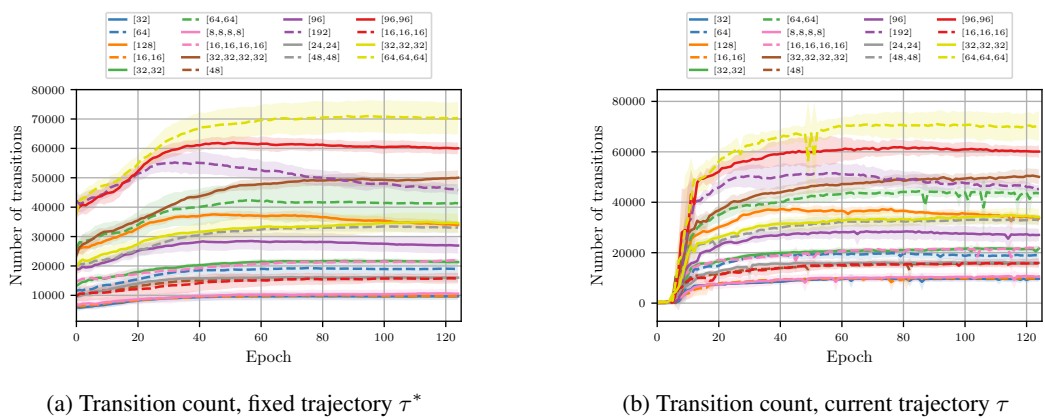

(a) Transition count, fixed trajectory $\tau^*$

(b) Transition count, current trajectory $\tau$

Figure 40: Evolution of the number of transitions over fixed and current trajectories during training for all policy network architectures trained on HalfCheetah. This figure is included for the sake of completeness.