# OpenReview forum: "Understanding the Evolution of Linear Regions in Deep Reinforcement Learning"
_NeurIPS.cc/2022/Conference — NeurIPS 2022 Accept_

### Official Review · Reviewer_xjsV · 2022-06-23

**Rating:** 5
**Confidence:** 3
**Soundness:** 3 good
**Presentation:** 3 good
**Contribution:** 2 fair

**Summary:**

This paper studies the way neural network policies with RELU activations partition their input space into linear regions. This has been studied before in supervised learning settings, but not for RL policies yet. The authors study 4 continuous control tasks. The main empirical observation is that the complexity of the policies comes from fine-tuning their contribution, not in a significant growth in their number along trajectories of the final policy. The number of regions seems mostly related to the total number of neurons, a finding also observed in supervised learning.

**Questions:**

I do not fully understand what I need to learn from the “region densities”, or what you actually compute here. As I understand, you take the total visitation counts per visited region, divide these over (N * L(tau)), and then you average this number? Doesn’t this really just estatimate the number of unique regions you visited?
* For example, imagine I visited 4 regions, with counts [100,200,300,400], and assume N=10 and L(tau)=10. This would give densities [1,2,3,4], and therefore average density 2.5.
* Now assume I instead observed counts [200,200,300,300]. This would give densities [2,2,3,3], and therefore still average density 2.5. Whatever way I distribute the counts over the regions, if the total number of regions stays the same, the average density will as well.
* I think what I would be really interested in is the total number of uniquely visited regions, if possible in a smaller problem with a density-coloured map (instead of averaged over all regions, which throws away a lot of information).




**Limitations:**

Conclusion: This is a decent paper. I like that the authors focus on a conceptual/insight topic in RL, which is often missing. On the other hand, I do think these results are close to what was already known from supervised learning, and I slightly miss the motivation of the authors why things could be different in RL (nevertheless, always worth investigating). I am also a bit in doubt about the way the results are presented, especially regarding the density estimation: this could maybe be clearer.


**Strengths And Weaknesses:**

Strong:
* I think conceptual insight studies are always relevant. As the authors mention as well, in RL we usually only look at the learning curves, which is a pity and probably misses much relevant insight.
* The overall paper is clearly written and easy to follow.
* I like Figure 3 (although you could maybe separately draw the Toy environment), it provides good insight.

Weak:
* See my comment in the Questions section.
* The authors only study four low-dimensional continuous control tasks, which arguably does not give a lot of variety. I therefore think the conclusions are a bit strong. I would have much preferred if there were 1 or 2 tasks from the Mujoco set (with indeed many cycles), but also some other types of tasks to validate and broader the scope of the conclusions. I think the current conclusions need to be downplayed a bit.
* The results confirm what we already knew from similar analysis in supervised learning. Of course, one still has to try in the RL setting to confirm this, but I think the paper needs to discuss why we would expect things to be different in the RL case? (I would not a clear reason: RL is mostly a data generation and target computation problem, after that the NN training process is simply a supervising learning problem).

Smaller:
* Figure 2 needs more explanation ( a better caption) to understand what is going on.
* I printed your version and all references were invisible (due to the very light green color, I would pick something darker).

---

> ### Author Response · Authors · 2022-08-02
> **Response to reviewer xjsV**
>
> ### The authors only study four low-dimensional continuous control tasks, which arguably does not give a lot of variety. I therefore think the conclusions are a bit strong. I would have much preferred if there were 1 or 2 tasks from the Mujoco set (with indeed many cycles), but also some other types of tasks to validate and broader the scope of the conclusions. I think the current conclusions need to be downplayed a bit.
> Please refer to our overall response  (“Choice of Environment”) for a discussion on tasks considered here.
> ### The results confirm what we already knew from a similar analysis in supervised learning. Of course, one still has to try in the RL setting to confirm this, but I think the paper needs to discuss why we would expect things to be different in the RL case?
> Please refer to our overall response for a full discussion on how RL setting is different from the supervised learning setting.
> ### Figure 2 needs more explanation ( a better caption) to understand what is going on.
> Figure 2 provides a visual explanation of activation patterns and linear regions, and lines 156-161 provide an explanation about it. Regardless, we did update the caption of figure 2 to provide a more clear description of the figure. Please refer to our revised submission.
> ### I printed your version and all references were invisible (due to the very light green color, I would pick something darker).
> Thank you for pointing this out. The hyperref colors are now updated to a darker orange for better visibility.
> ### I do not fully understand what I need to learn from the “region densities”, or what you actually compute here.
> Thank you for this clarification question. “Region densities” are essentially the normalized total number of transitions over some trajectory. To clarify, to some trajectory $\tau$, we first compute the total number of transitions between regions as we sweep the state space of the policy over $\tau$. We call this metric $R_T(\tau)$. Now, the region density over $\tau$ will be $\rho(\tau) = R_T(\tau) / (N L(\tau))$ where $N$ is the number of hidden units in the policy network and $L(\tau)$ indicates the length of trajectory $\tau$.
> In addition to region densities, we compute the number of unique regions trajectory $\tau$ will visit and call it $R_U(\tau)$. We do not plot this metric directly but we instead plot $R_T(\tau) - R_U(\tau)$ as “the number of repeat visits to regions” (please refer to Figure 4.f). Therefore, the number of unique visited regions can be infrared from region density plots and number of repeat visits to regions plots.
> The reason why we are mostly interested in region densities (number of transitions) instead of the number of unique visited regions is that the former captures the information about how the policy actually views the state space.
> ### For example, imagine I visited 4 regions, with counts [100,200,300,400], and assume N=10 and L(tau)=10. This would give densities [1,2,3,4], and therefore average density of 2.5.
> Continuing the discussion from the above question, $R_U(\tau) = 4$ for your example. Assuming each count in your example shows the number of times you visit each region, $R_T(\tau)$ will depend on the order you visit the regions. If you visit the regions one at a time, switching between different regions only once (exiting previous region to enter the next), then $R_T(\tau) = R_U(\tau) - 1 = 3$. This way, the density $\rho(\tau) = R_T(\tau) / (N L(\tau)) = 0.03$. You can imagine, by jumping back-and-forth between different regions, the number of transitions would increase and result in a higher $R_T(\tau)$ and a higher density. This example actually shows why density (number of transitions) can be a more informative metric than the number of unique visited regions for our case of study.

---

> ### Author Response · Authors · 2022-08-09
> **Reminder - discussion ends in less than 12 hours**
>
> We would like to kindly remind you that the discussion window will close in less than 12 hours. We will highly appreciate it if you could give us your thoughts on our responses as well as the additional experiments we provided. Your feedback is very valuable for us. We would also be more than happy to continue the discussion if you have any further questions or concerns.

---

### Official Review · Reviewer_yuJc · 2022-07-12

**Rating:** 5
**Confidence:** 4
**Soundness:** 3 good
**Presentation:** 3 good
**Contribution:** 2 fair

**Summary:**

Summary
-------

To better understand (relu-based) neural network policies in RL, this
paper proposes to measure the counts and densities of linear regions
during training. Using continuous control (MuJoCo) environments, the
authors empirically measure the linear regions' visitations (and other
quantities) while training PPO. They find that the evolution of the
policy's linear regions can be seen to increase for fixed trajectory
lengths, in accordance with results in supervised learning. For
on-policy and current trajectories, however, the density count decreases
during training.





**Questions:**


Detailed Comments
-----------------

-   Line 25-26: Some environments do allow for interpreting changes to
    policy over time. This is usually done for tabular environments
    however, such as gridworlds, and at the state-level rather than
    activation-level as in your submission. See, for example, Pardo et.
    al (2018), "Time Limits in RL".
-   Line 166: Is it more informative to count regions that are not
    explicitly encountered? This seems like it would lead to an
    overestimate.
-   Line 169-184: I think the ideas here would be better described in a
    figure/diagram or even an algorithm, rather than in a paragraph.
-   Line 185: Is there a specific reason for using the final policy as
    the "fixed" trajectory? This would presumably be the longest
    trajectory, but what exactly is a "fully-trained" policy.
-   Line 219: Fixed final trajectory. The policy is fixed, but how is
    the trajectory fixed? The policy itself is stochastic, so the
    trajectory cannot be fixed unless you take only the greedy action.
    Even then, there will be some variation in the start state that
    introduces variation in the trajectory.
-   Figure 4b: how is the change in policy measured along a fixed
    trajectory? The x-axis goes over epochs, which suggest that this is
    measured during training. Is the "fixed trajectory" fixed before
    training?
-   Figure 5: Is there any expectation that this coarse grained (only 3
    points) visualization should show significant differences? I think
    this may be more clear in a truly 2 dimensional problem, such as an
    LQR.
-   Line 243: It is not clear that revisiting a linear region in the
    neural network is the same as revisiting states, i.e. exploration.

Minor Comments
--------------

-   Figure 1: Minor, but can you clarify whether this this schematic
    isrepresenatitive of an actual problem? The 4th and 5th dot occur in
    the same region and so, should have the same action. And yet the
    transitions to the next dot look quite different. If this is just an
    illustration, a clarification would help

**Limitations:**

The limitations of the work are discussed in some detail in the last few paragraphs. The authors also acknowledge that algorithmic developments can be used for both good and bad, but that this paper does not necessarily contribute to an area that requires ethics review.

**Strengths And Weaknesses:**

Decision
--------

While I find this paper very interesting in the problem it investigates,
it is currently below the acceptance threshold. The main problem is that
the empirical investigation, while interesting, lacks a depth of
analysis (especially relative to the paper that it builds upon). A
theoretical analysis of the learning dynamics, while outside of the
scope of this paper, would also serve to strengthen its contribution.
Another thing that would help is to better characterize the difference
between RL and supervised learning (this is somewhat addressed in the
trajectory lengths, but the larger issue of temporality and changing
state distribution is missing). For this paper to meet the acceptance
threshold, it would need either: a deeper analysis on perhaps a simpler
problem, a broader analysis on more RL algorithms, or a theoretical
study. These changes, however, are outside the scope of a revision.

Strengths
---------

-   Interesting and novel investigation on the structure of relu-based
    networks in on-policy and continuous control.
-   Positioned well with respect to previous work in this area,
    especially in supervised learning.

Weaknesses
----------

-   Results, while interesting, do not shed much light on the
    peculiarities of RL policies. There are many differences between
    supervised learning and RL that is not well reflected in the problem
    being studied (namely PPO). There are many environments invesigated
    (in the appendix, at least). But, for an experimental paper, I would
    expect a more thorough evaluation using more algorithms or some
    specific hypothesis/justification for the limitation to PPO.
-   Lack of theory relative to other work in this area. Compared to
    supervised learning, this is obviously very difficult to do in RL.
    With the lack of "exciting" empirical results, however, theory could
    fill in this gap by providing the framework for thinking about
    relu-based neural networks and their use in RL.

---

> ### Author Response · Authors · 2022-08-02
> **Response to reviewer yuJc**
>
> ### Results, while interesting, do not shed much light on the peculiarities of RL policies.
> Please refer to our overall response for a discussion related to this point.
>
> ### Lack of theory relative to other work in this area.
> Please refer to our overall response  (“Theory”).
>
> ### Line 25-26: Some environments do allow for interpreting changes to policy over time. This is usually done for tabular environments however, such as gridworlds, and at the state-level rather than activation-level as in your submission. See, for example, Pardo et. al (2018), "Time Limits in RL".
> Good point.
>
> ### Line 166: Is it more informative to count regions that are not explicitly encountered? This seems like it would lead to an overestimate.
> This is to ensure that we do not miss very small regions. This is especially important in higher dimensional environments where many small regions exist within two consecutive steps on a policy trajectory. In many continuous RL settings, the control frequency can be increased without a significant impact on the resulting trajectories.
>
> ### Line 169-184: I think the ideas here would be better described in a figure/diagram or even an algorithm, rather than in a paragraph.
> Thanks, good idea;  we plan to add a diagram to the supplemental material.
>
> ### Line 185: Is there a specific reason for using the final policy as the "fixed" trajectory? This would presumably be the longest trajectory, but what exactly is a "fully-trained" policy.
> To clarify, we save a number of snapshots from the policy network as it is being trained in iterations. Once we converge to a final policy (that achieves state-of-the-art performance on its specific task; i.e. is “fully-trained”), we go back to these snapshots and evaluate them with respect to the final policy snapshot. More particularly, we sample a trajectory ($\tau^*$) from the final policy snapshot, iterate over all saved snapshots and compute the metrics over the state space of these snapshots along the fixed trajectory $\tau^*$. We call $\tau^*$ the “fixed trajectory” as it remains the same for all snapshots of the policy (compared with the choice of sampling the trajectory from the same snapshot that is being evaluated which we denote by $\tau$ in this work). The use of fixed trajectory offers a direct picture of the linear-region density along a meaningful, fixed region of the state space during training, i.e., that of the final optimized policy.
>
> ### Line 219: Fixed final trajectory. The policy is fixed, but how is the trajectory fixed? The policy itself is stochastic, so the trajectory cannot be fixed unless you take only the greedy action. Even then, there will be some variation in the start state that introduces variation in the trajectory.
> Please refer to the answer to the above question.
>
> ### Figure 4b: how is the change in policy measured along a fixed trajectory? The x-axis goes over epochs, which suggest that this is measured during training. Is the "fixed trajectory" fixed before training?
> Please refer to the answer to the question on line 185.
>
> ### Figure 5: Is there any expectation that this coarse grained (only 3 points) visualization should show significant differences? I think this may be more clear in a truly 2-dimensional problem, such as an LQR.
> These diagrams have been proposed in the context of even higher-dimensional inputs, i.e., images, where in some cases there have been visible differences in density.  We thus choose to also use this same type of visualization.
>
> ### Line 243: It is not clear that revisiting a linear region in the neural network is the same as revisiting states, i.e. exploration.
> True.  In continuous state, continuous action settings, we never truly revisit exactly the same state in any case.
> ### Figure 1: Minor, but can you clarify whether this this schematic isrepresenatitive of an actual problem? The 4th and 5th dot occur in the same region and so, should have the same action. And yet the transitions to the next dot look quite different. If this is just an illustration, a clarification would help.
> Indeed, Figure 1 is a schematic hand drawn. But in general, regions correspond to areas of the state space where the output is a linear function of the input, and thus not constant-valued. Therefore, we do not expect to see the same actions within a linear region.

---

> > ### Comment · Reviewer_yuJc · 2022-08-08
> > **Shared reply and individual response helpful**
> >
> > Thank you for clarifying these points. The shared reply, in particular, was helpful in contextualizing the paper. As I said before, this work is interesting and has potential to ellucidate better understanding of the policies learned by neural networks and different RL algorithms.
> >
> > Regarding your reply to my confusion in line 185: I understand a choice needs to be made in terms of defining a trajectory, and the trajectory of the converged policy is a sensible choice. Some confusion still remains as to the signifiance of choosing a particular realization of a policy's trajectory. As a though experiment, consider two different but "converged" policies (for example, different gaits in a mujoco walking env) that realize the same return. Is it necessarily the case that the linear-region metrics on one trajectory, with respect to one policy, will necessarily hold for the other trajectory? If the trajectories are "non-intersecting" then the state distribution will be quite different. A particular policy may not be performant on the trajectory of the other policy, which may make conclusions regarding the linear regions encountered not reflective of the linear regions on the on-policy trajectory.
> >
> > While the example I describe above is a bit contrived, it is not clear that the linear regions of a particular snapshot on the "fixed trajectory" should be reflective of the linear regions on the on-policy trajectory. These difficulties, among others like temporality and trajectory lengths, are unique to RL, but are not prominently featured in this submission. This is why I suggested that a deeper analysis on a simpler problem (LQR, for example) may help clarify these details, before providing the results on more realistic environments.
> >
> > All that being said, I do think there is merit to this work. The revision, shared reply, other reviews and discussions have made me consider this contribution in a more positive light. Although I still have reservations, specifically on the points listed above and the overall significance of the contribution, I will raise my score to a 5 to reflect the improvements to the submission.

---

> > > ### Author Response · Authors · 2022-08-09
> > > **Re: response**
> > >
> > > We appreciate the time you spent reviewing our updates, reading our responses, and giving us valuable feedback. We are glad that you found our overall and individual responses useful.
> > >
> > > ### The choice of trajectories.
> > > One of the key questions we were interested in answering in this work was **whether the density of linear regions along trajectories arising from the policy increases with training in order to afford better control**. As you mention, we need to define some trajectory from the policy for our evaluations. We tried two choices of trajectories: 1) a trajectory sampled from the final trained policy ($\tau^*$) as a “fixed” meaningful region of the state space and, 2) a trajectory sampled from the current snapshot of the policy ($\tau$) as the actual parts of the state space the policy visits during training. Our observations for policies trained with PPO on the four continuous control tasks showed: 1) a moderate increase (within 50%) in region density over $\tau^*$ during training and, 2) a decrease in region density over $\tau$ during training. Therefore, the surprising finding here is that **the complexity of RL policies does not come principally from increased density on-and-around the optimal trajectories but rather from fine-tuning their contribution**.
> > >
> > > The question on the choice of trajectories motivates us to further think about other potentially meaningful trajectories we can study for counting regions of a deep RL policy. So, thank you for bringing this up. Your question on *whether we would see the same results if we evaluate two different but "converged" policies on the same task but over a fixed trajectory that comes from one of the policies* is also very interesting. Doing an experiment like this would tell us whether the state space divisions are the same between the two such policies since we are looking at the same region of the state space of both the policies (trajectory sampled from one of the policies). However, in this work, our main focus is on answering how the metrics evolve during training a single policy and we believe that our experiments on the fixed trajectory and current trajectory are a means of answering this question.

---

### Official Review · Reviewer_5UGR · 2022-07-12

**Rating:** 6
**Confidence:** 4
**Soundness:** 3 good
**Presentation:** 3 good
**Contribution:** 3 good

**Summary:**

The authors aim to describe the evolution of the linear regions that takes place in a policy parameterized by a neural network while being optimized via reinforcement learning. ReLU activations make calculating these regions straightforward, but until now these sorts of analyses have been limited to supervised learning. Reinforcement learning is different in that it involves trajectories, which makes questions around region densities more meaningful (don't need to interpolate between IID datapoints).

The author investigate the effect of depth on the number of unique regions as well as how these regions evolve over the course of learning. Both of these questions are complicated by the choice of trajectory distributions; the authors do these analyses for both the current trajectory distribution as well as that induced by the final (near optimal) policy.

Experiments are conducted across 4 locomotion tasks, with policies optimized via PPO. They show that while the final trajectories have their region density increase over time, the region density of the current policy's trajectories actually decrease after a large initial spike.

**Questions:**

1) Would these results hold for other RL algorithms and environments?

(The below questions are more speculative, I don't expect you to run more experiments or anything)

2) How would these results differ for behavior cloning? Do you think this is more more sequence-learning or RL specifically?

3) Would more intelligent (or even optimal) exploration result in current-policy region density increasing monotonically?

**Limitations:**

Yes.

**Strengths And Weaknesses:**

# Strengths

1) This is quite a detail-oriented paper, so it's to the authors' credit that I was able to follow along quite easily. The writing quality was great and the related-works section in particular headed off many of my initial questions about this work's relationship to the rest of the field.

2)The high level of idea of bringing analyses developed for supervised deep learning into DRL is well-worth pursuing, particularly given the points made about trajectory-based region density estimation being more natural than in the IID setting.

3) The effect sizes are very strong and quite consistent across architectures. The follow-up experiment on repeat visits to regions is compelling evidence in favor of explanation of the surprising result that region density decreases over time for the current policy (i.e. it's due to poor initial exploration).

# Weaknesses

1) Having a single figure in the main paper that aggregates all of the key results across environments is sorely needed. It's not quite obvious to me how to do this since the y-axis scales might be significantly different, but I'm sure some sort of normalization scheme could allow us to see that the shape of the results are similar across environments.

2) I'm not confident that these results will generalize across environments, which limits the potential impact of this work. To the authors' credit, they acknowledge that focusing on locomotion environments is limiting, but that begs the question as to why they imposed this limitation. I believe the code-base you're using could just as easily be applied to e.g. manipulator environments, and doing so would build confidence in the applicability of your results.

3) Similarly, picking a single RL algorithm feels limiting. Something purely value-based (e.g. Q-learning) would be particularly interesting, as value-space is very differently behaved compared to policy-space. Indeed, even just redoing your analyses on the value network of your PPO agents would be quite interesting.

4) Figure 5 is very pretty, but I'm not sure what it is supposed to be communicating.

5) Some more motivation about why we should care about the outcome of Q1 and Q2 would be nice.

---

> ### Author Response · Authors · 2022-08-02
> **Response to reviewer 5UGR**
>
> ### Having a single figure in the main paper that aggregates all of the key results across environments is sorely needed.
> We agree that being able to visualize the key findings across all the environments would help convey the message of our work more clearly. However, because of the large differences between the metric scales across different tasks, we decided to show the pattern of observations, first for one reference environment (HalfCheetah) in Figure 4. Then, for each metric, we provide the plots for all environments, to provide a means of comparison between new environments’ results, and the reference environment. We have kept the same style for the supplementary material plots as well.
>
> ### I'm not confident that these results will generalize across environments, which limits the potential impact of this work.
> We provide a short discussion about the choice of environments in our overall response; please refer to "Choice of Environment". We have repeated our experiments on one additional task of LunarLanderContinuous. This environment is interesting as it does not have the cyclic nature of the other locomotion tasks and thus, allows us to investigate our hypotheses on the connection between some of the observed patterns and the cyclic nature of the tasks. The core finding is that the only discrepancy between the results of the locomotion tasks and LunarLander is in the density over current trajectories (and the length of the current trajectories).
> For LunarLander, observed density over current trajectories does not decrease during training and has the same range of values as the density of fixed and random-action trajectories. We argue that as previously noted in the paper, for the four locomotion tasks, the length of the cyclic trajectories increased during training. Whereas for LunarLander, the trajectory length initially increases, then plateaus once the agent converges. We have added the section “Decision Regions for a Non-Cyclic Task: LunarLander” in supplementary materials with the details, along with a brief discussion.
>
> ### Similarly, picking a single RL algorithm feels limiting.
> Please refer to “Choice of RL Algorithm” in the overall response and the supplementary material for full discussion and results.
>
> ### Something purely value-based (e.g. Q-learning) would be particularly interesting, as value-space is very differently behaved compared to policy-space. Indeed, even just redoing your analyses on the value network of your PPO agents would be quite interesting.
> We have now included experiments on the value network of PPO policies. The core finding is that results are consistent between the value network and policy network regions, i.e., we observe a slight increase in the region densities as measured along a fixed final-policy trajectory. We have added the section “Value Network Analysis” in supplementary materials with the details. Please refer to the supplementary material to view the results and the full discussion.
>
> ### Some more motivation about why we should care about the outcome of Q1 and Q2 would be nice.
> Please refer to the overall response where we outline the relevance and importance of understanding the evolution of linear regions in deep RL.
>
> ### Would these results hold for other RL algorithms and environments?
> We now include additional results to better address this question; please refer to the overall response  ("Choice of RL Algorithm (PPO)" and “Choice of Environment”) as well as the revised draft, which includes results for SAC, as well as an acyclic task of LunarLander.
>
> ### How would these results differ for behavior cloning? Do you think this is more sequence-learning or RL specifically?
> We share your interest in this question. Our overall response (5) alludes to the importance of examining the path of non-IID learning. The policy's "learning history" plays a significant role in our results, as early trajectories inform the template cell divisions that later evolve with further training. We do not yet have results available for behavior cloning but will include these when available.
>
> ### Would more intelligent (or even optimal) exploration result in current-policy region density increasing monotonically?
> This is an interesting question, for which we do not currently have a sound answer.

---

> > ### Comment · Reviewer_5UGR · 2022-08-08
> > **Thank you for your response**
> >
> > I particularly appreciate the additional experiments and summary plots.
> >
> > That said, it is a bit unfortunate that so much changes based on the RL algorithm and task. I viewed the decreasing region density of the current policy's trajectories as the most prominent findings, but this effect disappears for the Lunar Lander task. Similarly, the qualitatively different behavior shown between PPO and SAC in Figure 7 (Appendix) shakes my confidence in the generality of your conclusions.
> >
> > I'm going to leave my score at a 6. The additional work is commendable, but the results shine light on the underlying specificity of the results. But this is still a good paper for raising the questions that it does and I could see these initial experimental results leading to more definitive work in the future, so I'm recommending its publication.

---

> > > ### Author Response · Authors · 2022-08-09
> > > **Re: response**
> > >
> > > We appreciate the time you spent reviewing our additional experiments, reading our responses, and giving us valuable feedback. We are glad that you found our additional experiments useful.
> > >
> > > ### Changes based on the RL algorithm and task.
> > > One of the key questions we sought to find answers to in this work was **whether the density of linear regions along trajectories arising from the policy increases with training in order to afford better control**. We tried two choices of trajectories from the policy: 1) a trajectory sampled from the final trained policy ($\tau^*$) and, 2) a trajectory sampled from the current snapshot of the policy ($\tau$). Our observations for policies trained with PPO on the four continuous control tasks showed: 1) a moderate increase (within 50%) in region density over $\tau^*$ during training and, 2) a decrease in region density over $\tau$ during training. Therefore, the surprising finding, for our experimental setup, is that **the complexity of RL policies does not come principally from increased density on-and-around the optimal trajectories but rather from fine-tuning their contribution**.
> > >
> > > Please note that although the evolution of the region density over current trajectories is an interesting metric to study, it is directly affected by how the length of trajectories evolves during training themselves. As discussed in the paper, the length of cyclic trajectories for the locomotion tasks increases during training such that it causes a drop in the region density as viewed by these trajectories. However, we do not expect the same property to hold for non-cyclic tasks as well. Therefore, the discrepancy between the results for locomotion tasks and lunarlander which is in the evolution of the density over current trajectories **is not in contrast with the rest of our results**. Moreover, the density over the fixed, current, and random-action trajectories for lunarlander are within the same range similar to the rest of tasks which is in alignment with the overall story of the paper. Our explanation of this observation is available in section 5.3 and the supplementary materials section.
> > >
> > > Regarding the difference between the results of PPO and SAC, we like to note that the only difference between the two sets of results is in the evolution of the density **over fixed trajectories**. After reviewing the first paragraph of section 5.3, we realized we have not clarified that the density we are referring to is the density over fixed trajectories, which may have caused confusion. We will reword this in our new revision. To clarify, in PPO, a moderate increase in the density over fixed trajectories is observed while with SAC the density over fixed trajectories is initially high then rapidly drops, and then rises again. Although the evolution pattern is different, by looking at the full sets of results for SAC, we can see that the density over fixed trajectory, current trajectories, and random-action trajectories **are within the same range**. This again **supports our main finding which is the complexity of RL policies is not principally captured by increasing the granularity of linear regions over trajectories from the policies**. We hypothesize that the observation of fixed trajectory region density is due to a combination of the entropy bonus for SAC which anneals away during training as well as the different network initializations it uses. The complete discussion on this is provided in section 5.3 and the supplementary material section.

---

### Official Review · Reviewer_zgAD · 2022-07-13

**Rating:** 5
**Confidence:** 2
**Soundness:** 3 good
**Presentation:** 3 good
**Contribution:** 2 fair

**Summary:**

The paper attempts to understand how the activations of the agent’s policies (represented as a neural network) evolves during training on standard continuous control tasks. The hypothesis is that the number of regions partitioned (i.e., density) increases in areas where the policy visits often in order to afford finer control for the learning agent in those regions. The paper uses recent theoretical results and empirical observations from supervised learning literature to draw conclusions in RL.


**Questions:**

* The paper and method used to evaluate the neuron activations in a policy network are interesting. But it is hard to understand how to use those observations in informing future research in RL. A discussion on why these findings are important to RL or more generally in Deep RL would be valuable.
* Were the linear regions different from the earlier layers to the later layers of the policy network? Were they task specific or agnostic to the considered tasks?
* Can we expect similar observations to emerge on agents like DQN?


**Limitations:**

The paper presents an interesting analysis of existing policy learning algorithms in terms of its neuron activations. But it is unclear on how to apply these findings in future research.


**Strengths And Weaknesses:**

Strengths:
* The paper is well-presented and the results are interesting. It is an analysis method to understand the RL agents.
* Empirical results delve into how the activations within the neural network evolve with training on standard RL tasks.


Weaknesses:
* It is unclear as to how the observations made in this paper inform future research. A discussion around this would be much appreciated.

---

> ### Author Response · Authors · 2022-08-02
> **Response to reviewer zgAD**
>
> ### A discussion on why these findings are important to RL or more generally in Deep RL would be valuable.
> We outline potential future research avenues in our overall response, and now also in the paper.
>
> ### Were the linear regions different from the earlier layers to the later layers of the policy network? Were they task specific or agnostic to the considered tasks?
> Linear regions induced by each layer of the policy network functionally create finer divisions. In that sense, there is a distinction between earlier and later layers as later layers introduce "edits" to the existing linear regions of the earlier layers. Particularly, each neuron in the first layer defines a linear boundary that partitions the input space into two regions. Neurons in the second layer combine and split these linear boundaries into higher level patterns of regions, and so on. One may view this as each additional layer introducing higher-frequency information required for finer cell division.
>
> ### Can we expect similar observations to emerge on agents like DQN?
> Our experiments were limited to tasks involving continuous states and actions and empirical study of the policy network. Basic DQN uses a maxQ instead of a policy network. However, for DQN, decision regions could still be studied for the value function network. We now also study this, for our original tasks.

---

> ### Author Response · Authors · 2022-08-09
> **Reminder - discussion ends in less than 12 hours**
>
> We would like to kindly remind you that the discussion window will close in less than 12 hours. We would highly appreciate it if you could give us your thoughts on our responses as well as the additional experiments we provided. Your feedback is very valuable for us. We would also be more than happy to continue the discussion if you have any further questions or concerns.

---

### Author Response · Authors · 2022-08-02
**Overall response (1/2)**

Thank you for the detailed reviews, related feedback, and suggestions. We are glad to see the strong interest in this problem and the appreciation for the clarity of the writing.

We have now further added results for SAC, an additional non-cyclical task (LunarLander), and studied linear-region evolution in the value function.  Thank you for suggesting these directions, as we believe they have added significant value.

Several readers wished to hear more on why understanding the evolution of linear regions in deep RL is a relevant and important problem.  Below we list a number of reasons, and we will do a better job of (re)articulating these in the paper.

1. The continuous state trajectories resulting from continuous control problems offer an __inherently-meaningful path__ along which to measure region densities.  In contrast, the lines currently used for studying region density in the supervised learning (SL) setting represent linear interpolations between the pixel values of two MNIST images. This is not always meaningful, e.g., producing composite images with ghosting.

2. Existing studies of linear regions in the supervised learning (SL) setting are not fully self-consistent in their conclusions. Thus it is even less clear how these results might carry over to the RL policy setting.  For example, [Novak et al. 2018] present evidence for lower linear-region density in the vicinity of data points (see Fig 2 therein), whereas [Hanin et al., 2019a] find a qualitatively different result (see caption of Fig 8).

3. While the argument in SL for image classification is that lower decision-region density around data-points is beneficial for generalization, e.g., [Novak et al. 2018], lower densities around data-points is arguably bad from a control perspective, where we would wish to see more expressivity (higher densities) around states frequented by the policy, to allow for fine-grained control. So what actually happens in the RL setting?  Is the region density higher or lower than average along the trajectory visited by a learned policy? If higher density, does this density slowly track the shifting state distribution observed during RL training? We seek to answer such questions.

4. Piecewise affine (“linear”) control strategies are commonly designed into control systems, i.e., via gain scheduling and switched linear dynamical systems. Deep RL with ReLU-based policies provides learned versions of these. Thus understanding how these regions are designed and distributed by deep RL builds bridges to control methods.

5. The evolution of the linear decision regions in non-IID settings is not currently understood. The continuous states-and-actions RL setting provides a meaningful and grounded setting for exploring this. Learning results are already known to be learning-path dependent via the known impacts of network initialization, curriculum learning, and catastrophic forgetting. Our current work does not tackle the general problem of how a wide range of IID settings impacts linear-region evolution. However, we make progress for a setting that many people care about, i.e., deep RL.

6.  In the SL setting, the number of neurons is the principal factor that impacts policy region density, and the policy layer depth has a minimal impact.  Arguably, the RL community currently does not yet think about policy architecture design in these terms.  Our results show that RL policy expressivity (as measured via densities) is also mainly a function of the number of neurons.

---

> ### Author Response · Authors · 2022-08-02
> **Overall response (2/2)**
>
> ### Impact on Future RL research
>
> Our work focuses on “understanding” via an empirical approach, and we believe this is important on its own. However, we envisage various future directions:
> (i) understanding the impact of non-IID data on decision regions more generally; (ii) consideration of RL algorithms that have less “learning path dependence”, e.g., via frequent (re)distillation onto randomly reinitialized policies; (iii) using visited regions to efficiently distill more interpretable non-parametric policies;  (iv) in real-world control settings, using knowledge of decision regions, their sizes, and their orderings to better understand exploration and safety issues.
>
>
> ### Choice of Environment
>
> We have now also added results and discussion for a non-cyclic continuous task of LunarLanderContinuous. The core result is that the evolution of the region density over current trajectories is different than for the four locomotion tasks, which we attribute to the cyclic nature of the locomotion tasks. For the locomotion tasks, the length of cyclic trajectories grows during training, accounting for a decrease in the observed region density. In LunarLander, the length of trajectories plateaus after the agent converges and the observed density has a slow increase before it plateaus.
>
> More generally, the design space for our experiments is large; (region-measurement choices) x (18 policy network sizes) x (RL-tasks) x (RL algorithms) x (RL hyperparams). We focus on measurement and visualization choices, the impact of policy network sizes, and some of the most common RL continuous tasks. For a more tractable scope, we use one of the most common RL algorithms and justified RL hyperparameter choices. To the best of our knowledge, our work is the first analysis of the evolution of decision regions for RL settings, and thus we target the most salient aspects of the problem and relate it to existing work in the supervised learning setting.
>
>
> ### Choice of RL Algorithm
>
> Our work focuses on PPO, which is one of the most widely used RL algorithms, favored for its robustness and the available baseline implementations. We have now also added results and discussion for SAC. The core result is that the initial evolution of the region density is different than for PPO, which we attribute to a combination of the different initialization strategies (as per the baseline SAC implementation) and the evolution during training of the exploration bonus. The SAC setting provides a contrast with the SL setting in terms of the evolution of region densities over time.
>
>
> ### Theory
>
> The RL setting with continuous states and actions is particularly challenging for theoretical analysis, particularly when considering the need to leverage analytical tools for both the policy representation and the self-adapting state distributions produced by the RL algorithms. We are excited about potential progress in this direction, but leave it as future work. In the meantime, our work provides the community with the first understanding of what decision regions look like and how they evolve in ReLU-based RL policies.
>
>
> ___________________________
> ### Update
> We found a mistake in including figures for the initial set of additional experiments in the supplementary material submitted on August 2. We mistakenly replaced the first set of complementary results of PPO on the three remaining tasks (which was provided in the first supplementary material version submitted on May 26) with the new SAC results in the new supplementary material revision. Therefore, we replaced the complementary PPO plots with their original copies and submitted a new revision for the supplementary material.

---

### Meta-Review · Area_Chair_CwAm · 2022-08-24

**Recommendation:** Accept
**Confidence:** Less certain

**Metareview:**

This paper provides an empirical analysis of RL methods via the lens of linear regions induced by NN-based agents. All the reviewers agreed the paper was well written and it was wonderful to see an understanding based paper. The paper is clear about its limitations, very detail oriented & scientific, and borrows & checks the validity of an analysis technique from supervised learning. The authors nicely added an additional environment and agent to the paper at the reviewers request.

This paper is very close. Several reviewers asked in one way or another "where is the evidence that we should care about linear regions" and "how will this inform future research?". The majority of the reviewers found the response and updated paper did not clearly address these questions. Indeed, even the most positive reviewer pointed out that the 2 research questions posed in the introduction were not well motivated. This is concern #1.

Concern #2 regards the generality of the results: what trends we see across agents and environments. With the added experiments the trends were not clear. This can be ok especially when testing an idea shown to be important in SL. It can be useful to report such a result. However, in this case we might need a good variety of agents and environments (not necessarily large scale environments, just different ones).

In the end for such a paper to have impact we need either: (a) interesting trends to come out of the results and a clear articulated connection to future algorithmic development or (b) a clear body empirical results that say "yep we tried this and its not clear it helps". The current paper does a bit of both.

**Award:**

No

---

### Decision · Program_Chairs · 2022-09-14

Accept